# Endothelial pannexin-1 channels modulate macrophage and smooth muscle cell activation in abdominal aortic aneurysm formation

Amanda C. Filiberto [1], Michael D. Spinosa [2], Craig T. Elder [1], Gang Su[1], Victoria Leroy [1], Zachary Ladd [1], Guanyi Lu[1], J. Hunter Mehaffey[2], Morgan D. Salmon [2], Robert B. Hawkins[2], Kodi S. Ravichandran[3], Brant E. Isakson[4], Gilbert R. Upchurch Jr [1] & Ashish K. Sharma [1]✉

Pannexin-1 (Panx1) channels have been shown to regulate leukocyte trafficking and tissue inflammation but the mechanism of Panx1 in chronic vascular diseases like abdominal aortic aneurysms (AAA) is unknown. Here we demonstrate that Panx1 on endothelial cells, but not smooth muscle cells, orchestrate a cascade of signaling events to mediate vascular inflammation and remodeling. Mechanistically, Panx1 on endothelial cells acts as a conduit for ATP release that stimulates macrophage activation via P2X7 receptors and mitochondrial DNA release to increase IL-1β and HMGB1 secretion. Secondly, Panx1 signaling regulates smooth muscle cell-dependent intracellular $Ca^{2+}$ release and vascular remodeling via P2Y2 receptors. Panx1 blockade using probenecid markedly inhibits leukocyte transmigration, aortic inflammation and remodeling to mitigate AAA formation. Panx1 expression is upregulated in human AAAs and retrospective clinical data demonstrated reduced mortality in aortic aneurysm patients treated with Panx1 inhibitors. Collectively, these data identify Panx1 signaling as a contributory mechanism of AAA formation.

[1] Department of Surgery, University of Florida, Gainesville, FL, USA. [2] Department of Surgery, University of Virginia, Charlottesville, VA, USA. [3] Department of Microbiology, Immunology and Cancer Biology, University of Virginia, Charlottesville, VA, USA. [4] Department of Molecular Physiology and Biological Physics, University of Virginia, Charlottesville, VA, USA. ✉email: ashish.sharma@surgery.ufl.edu

bdominal aortic aneurysm (AAA) formation is primarily a vascular disease in the male elderly population, resulting in significant mortality[1–3]. The risk of aortic rupture is associated with increased size and rate of aortic-wall expansion, leading to a mortality of ~80%[4]. At present, there is no pharmacological therapy for AAA, with operative repair representing the only intervention that carries substantial morbidity[5]. The pathophysiology of AAA involves chronic inflammation, upregulation of proteolytic pathways, oxidative stress, and loss of arterial wall matrix[6,7]. The critical inflammatory pathways that lead to immune-cell and smooth-muscle-cell activation during AAA formation remain to be elucidated. Specifically, we focused on the previously unrecognized role of ion channels and transporters in the multifaceted crosstalk between resident aortic cells and infiltrating immune cells during aortic inflammation and remodeling in AAA formation.

Cellular communication of the intracellular compartment with the extracellular space through membrane channels has emerged as pivotal paracrine-signaling pathways involved in vascular homeostasis and disease[8–11]. Previous studies have shown that pannexin- and connexion-family channels can conduct ATP across the plasma membrane that can lead to leukocyte recruitment and regulate tissue inflammation[12,13]. There are three pannexin channels described in humans (PANX1–3), and several pathophysiological conditions have been linked to pannexin-1 channel regulation[14–18]. Panx1 isoforms oligomerize to form heptameric plasma-membrane channels that open during pathological conditions (i.e., ischemia and mechanical stress) to release ATP among other small signaling molecules[13,19]. The intracellular accumulation and subsequent release of extracellular ATP (eATP) can act as a danger-associated molecular-pattern (DAMP) molecule that can mediate paracrine signaling to upregulate intracellular communication. Immune-cell activation following Panx1-mediated eATP release can be mediated via various purinergic P2X and P2Y receptors to promote tissue inflammation. Our previous studies have characterized a key mediatory role of inflammasome activation and IL-1β secretion, as well as macrophage-dependent high mobility group box 1 (HMGB1) in regulation of aortic inflammation during AAA[20–22]. We have also defined that aortic smooth-muscle cell (SMC) apoptosis and activation can lead to an increase in matrix-metalloproteinase activity, leading to vascular remodeling during AAA formation. However, the contribution of aortic endothelial cells in this chronic inflammatory-disease process has not yet been deciphered.

Our hypothesis focused on the mechanism by which Panx1-mediated eATP signaling in vascular endothelium can play a major contributory role in macrophage and SMC activation, leukocyte trafficking, as well as cytokine milieu that culminates in vascular remodeling observed during AAA formation. Our results demonstrate that EC-specific Panx1, but not SMC Panx1, can regulate aortic diameter, inflammation and remodeling in AAA formation. Using two different murine models of AAA, we observe that pharmacological inhibition of Panx1 channels via probenecid leads to significant mitigation of vascular inflammation and remodeling, resulting in a protective phenotype. The mechanistic signaling of Panx1-specific eATP release from ECs regulates macrophage-dependent IL-1β and HMGB1 release via P2X7 receptors and mitochondrial DNA release. Moreover, eATP release from Panx1 channels from ECs also activates P2Y2 receptors and TRPV4 channels in SMCs, resulting in pro-inflammatory cytokine release and destabilization of intracellular $Ca^{2+}$ homeostasis that facilitates increased MMP2 activity and AAA formation. Finally, retrospective analysis of human AAA patient clinical data suggests a lower risk of mortality associated with probenecid and spironolactone that are known Panx1

inhibitors[12,23–26]. These data suggest aortic endothelial Panx1 contributing to AAA formation, and identify a previously unknown mechanism of aortic aneurysm pathobiology.

## Results

**Panx1 on endothelial cells regulates AAA formation and vascular inflammation.** Using the murine elastase AAA model, we investigated the role of Panx1 in EC- and SMC-specific knockout mice. Aortic diameter was significantly decreased in elastase-treated $EC$-$Panx1^{-/-}$, but not $SMC$-$Panx1^{-/-}$ mice, compared with elastase-treated WT mice on day 14 (72.1 ± 2.6% vs. 122.4 ± 1.8% and 113.7 ± 3.9%, respectively; Fig. 1a–c). There was no significant change in aortic diameter in WT mice treated with tamoxifen compared with deactivated elastase-treated mice (Supplementary Fig. 1). Pro-inflammatory cytokine expression (IL-1β, HMGB1, IFN-γ, IL-23, IL-17, MCP-1, MIP-1α, CXCL1, RANTES, and TNF-α) in aortic tissue was significantly attenuated in $EC$-$Panx1^{-/-}$, but not $SMC$-$Panx1^{-/-}$ mice, compared with elastase-treated WT mice (Fig. 1d–h). The aortic tissue content of ATP was significantly decreased on days 3, 7 and 14 in elastase-treated $EC$-$Panx1^{-/-}$ mice compared with elastase-treated WT mice and $SMC$-$Panx1^{-/-}$ mice (Fig. 1i). MMP2 activity was measured in aortic tissue and was significantly attenuated in elastase-treated $EC$-$Panx1^{-/-}$ compared with elastase-treated WT and $SMC$-$Panx1^{-/-}$ mice (Fig. 1j). Additionally, a marked increase in Panx1 expression was observed in aortic tissue of elastase-treated WT mice that was absent in elastase-treated $EC$-$Panx1^{-/-}$ mice (Fig. 1k). Elastase-treated $EC$-$Panx1^{-/-}$ mice demonstrated a significant decrease in immune-cell (PMNs, macrophages, and CD3 + T cell) infiltration and elastin-fiber disruption (VVG staining), as well as an increase in smooth-muscle-cell α-actin (SM-α-actin) expression compared with elastase-treated WT mice (Fig. 2a–f). No differences in immunostaining were observed in elastase-treated $SMC$-$Panx1^{-/-}$ compared with elastase-treated WT mice.

**Pharmacological inhibition of Panx1 mitigates AAA formation.** The elastase model of AAA was used to identify if treatment using a Panx1 inhibitor, probenecid (PBN), can mitigate AAA formation. PBN treatment attenuates aortic diameter in elastase-treated male WT mice compared with elastase treatment alone (61.4 ± 7.1% vs. 121.1 ± 3.9%, Fig. 3a–c). A marked decrease in immune cell (PMNs, macrophages, and CD3 + T cell) infiltration and elastin-fiber disruption (VVG staining), as well as an increase in SM-α-actin expression was observed in PBN-treated WT mice compared with elastase-treated WT mice (Fig. 3d–i). Similarly, pro-inflammatory cytokine expression (specifically IL-1β, HMGB1, and IL-17) in aortic tissue was significantly attenuated in elastase-treated WT mice administered with PBN compared with elastase-treated WT mice alone (Fig. 3j–n). The aortic tissue content of ATP was significantly decreased on day 14 in PBN-treated mice compared with elastase-treated WT mice (Fig. 3o). Plasma eATP levels were significantly mitigated in PBN-treated mice compared with untreated controls (Supplementary Fig. 2). Moreover, elastase-treated WT mice showed increased MMP2 activity compared with controls, and were attenuated after PBN treatment (Fig. 3p). Additionally, we investigated the protective role of PBN administered *after* the formation of AAAs. PBN treatment after AAA formation on day 7 significantly decreased aortic diameter in elastase-treated mice on day 14, suggesting a therapeutic and clinically relevant role for human translation (Supplementary Fig. 3).

The angiotensin-II/$ApoE^{-/-}$ model of AAA was used as a second model to confirm the protection observed by pharmacological inhibition of Panx1 channels. The mean aortic diameter

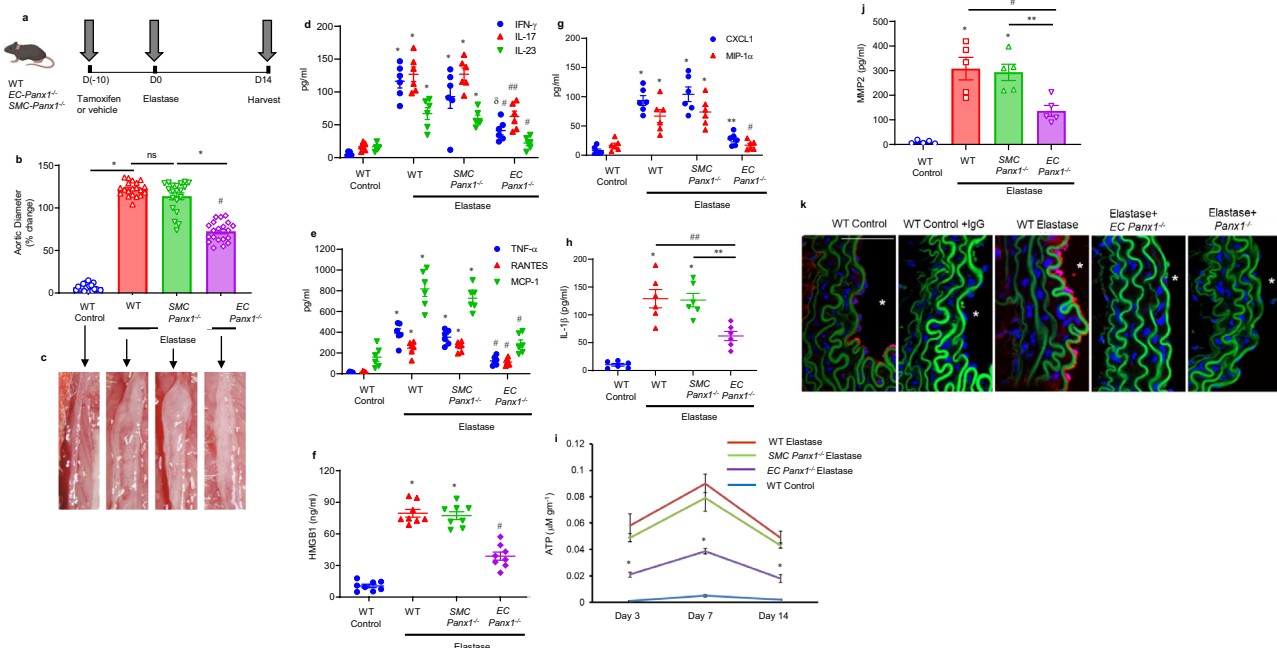

**Fig. 1 Panx1 on endothelial cells regulates AAA formation and vascular inflammation. a** Schematic description of the topical elastase-treatment model is shown. Inducible knockout male mice were treated with tamoxifen for 10 days prior to elastase treatment. WT or inducible knockout male mice were then treated with topical elastase, or heat-inactivated elastase (control), on day 0, and aortic diameter was measured on day 14. **b** Aortic diameter is significantly reduced in elastase-treated EC-Panx1$^{-/-}$, but not SMC-Panx1$^{-/-}$ mice, compared with elastase-treated WT mice. *$P < 0.0001$; #$P < 0.0001$ vs. WT elastase; ns, $P = 0.09$; $n = 20$ mice/group. **c** Representative images of aortic phenotype in all groups. **d–h** Pro-inflammatory cytokine expression in aortic tissue is significantly attenuated in elastase-treated EC-Panx1$^{-/-}$, but not SMC-Panx1$^{-/-}$ mice, compared with elastase-treated WT mice. *$P < 0.0001$ vs. WT control; #$P = 0.0004$ vs. WT elastase; δ$P = 0.01$ vs. SMC-Panx1$^{-/-}$ elastase; ##$P = 0.001$ vs. WT elastase and SMC-Panx1$^{-/-}$ elastase; **$P = 0.002$ vs. SMC-Panx1$^{-/-}$ elastase; $n = 6$/group. **i** Aortic tissue content of ATP was significantly attenuated in elastase-treated EC-Panx1$^{-/-}$, but not SMC-Panx1$^{-/-}$ mice, compared with elastase-treated WT mice on days 3, 7, and 14. *$P < 0.008$ vs WT elastase and SMC-Panx1$^{-/-}$ elastase; $n = 5$/group. **j** Expression of MMP2 in aortic tissue was significantly attenuated in elastase-treated EC-Panx1$^{-/-}$ mice compared with elastase-treated WT and SMC-Panx1$^{-/-}$ mice. *$P < 0.0001$ vs. WT control; #$P = 0.005$ vs. WT elastase and **$P = 0.01$ vs. SMC-Panx1$^{-/-}$ elastase; $n = 5$/group. All data above are represented as mean values ± SEM and comparative statistical analyses were done by one-way ANOVA followed by multiple comparisons. **k** Aortic tissue from elastase-treated WT mice demonstrated a marked increase in Panx1 expression on day 14 compared with elastase-treated EC-Panx1$^{-/-}$ mice. *: lumen; red: Panx1; green: elastic lamina and blue, blue: DAPI. Scale bar is 50 μm; $n = 5$/group. Representative images from independent experimental replicates are depicted.

was significantly attenuated in ApoE$^{-/-}$ mice treated with GSK2 compared with Ang II alone on day 28 ($0.78 \pm 0.03$ vs. $1.44 \pm 0.1$ mm, $P < 0.0001$, Fig. 4a–c). PBN-treated mice demonstrated significantly less infiltration of CD3 + T-cells, macrophages, and neutrophils, and decrease in elastic-fiber disruption as well as increase in SM α-actin expression compared with Ang-II treatment alone (Fig. 4d–i). A significant attenuation of inflammatory cytokines, including IL-1β and HMGB1, associated with aneurysm formation was seen in the PBN-treated mice compared with Ang II alone (Fig. 4j–n). The aortic tissue content of ATP was significantly decreased on day 28 in PBN-treated mice compared with Ang-II-treated ApoE$^{-/-}$ mice alone (Fig. 4o). Moreover, PBN-treated mice showed a decrease in MMP2 activity compared with untreated controls (Fig. 4p). Collectively, these results demonstrate that pharmacological inhibition of Panx1 can significantly mitigate aortic inflammation and vascular remodeling in AAA formation.

**Panx1-mediated ATP release by endothelial cells mediates neutrophil transmigration.** To delineate the Panx1-mediated release of eATP by ECs, transient elastase treatment of ECs was performed and eATP was measured at different time points. A significant multifold increase in eATP was observed by elastase-treated ECs compared with controls at 6- and 12 h (Fig. 5a). PBN-treated culture significantly attenuated elastase-induced eATP

release (Fig. 5a). Similarly, ECs were treated with elastase and/or cytomix (a combination of most relevant pro-inflammatory cytokines in AAA pathogenesis i.e., IL-1β + IL-17 + HMGB1), as we have previously shown[20,21,27]. eATP release was significantly attenuated by PBN treatment in elastase- and/or cytomix-treated ECs compared with untreated culture (Fig. 5b). These data demonstrate the ability of PBN to inhibit elastase and/or cytomix-induced eATP release by ECs.

To assess neutrophil transmigration through an endothelial monolayer after elastase or cytomix, we performed a neutrophil transendothelial migration assay using AECs and primary neutrophils isolated from WT mice (Fig. 5c). A significant increase in neutrophil transmigration occurred after exposure to elastase or cytomix, which was significantly attenuated by PBN treatment (Fig. 5d).

**EC-Panx1-mediated release of ATP modulates macrophage activation via P2X7 receptors.** To investigate the crosstalk between ECs and macrophages, we performed a conditioned media-transfer (CMT) experiment between ECs and macrophages (Fig. 5e). CMT from elastase-treated ECs to macrophages resulted in significant increase in IL-1β ($9.7 \pm 2.1$ vs. $0.6 \pm 0.2$ pg/ml, $p < 0.0001$, Fig. 5f) and HMGB1 ($6 \pm 1.1$ vs. $0.47 \pm 0.1$ ng/ml, $p < 0.001$, Fig. 5g) release compared with controls, respectively. CMT from ECs after pre-treatment with apyrase (10 U/ml), or

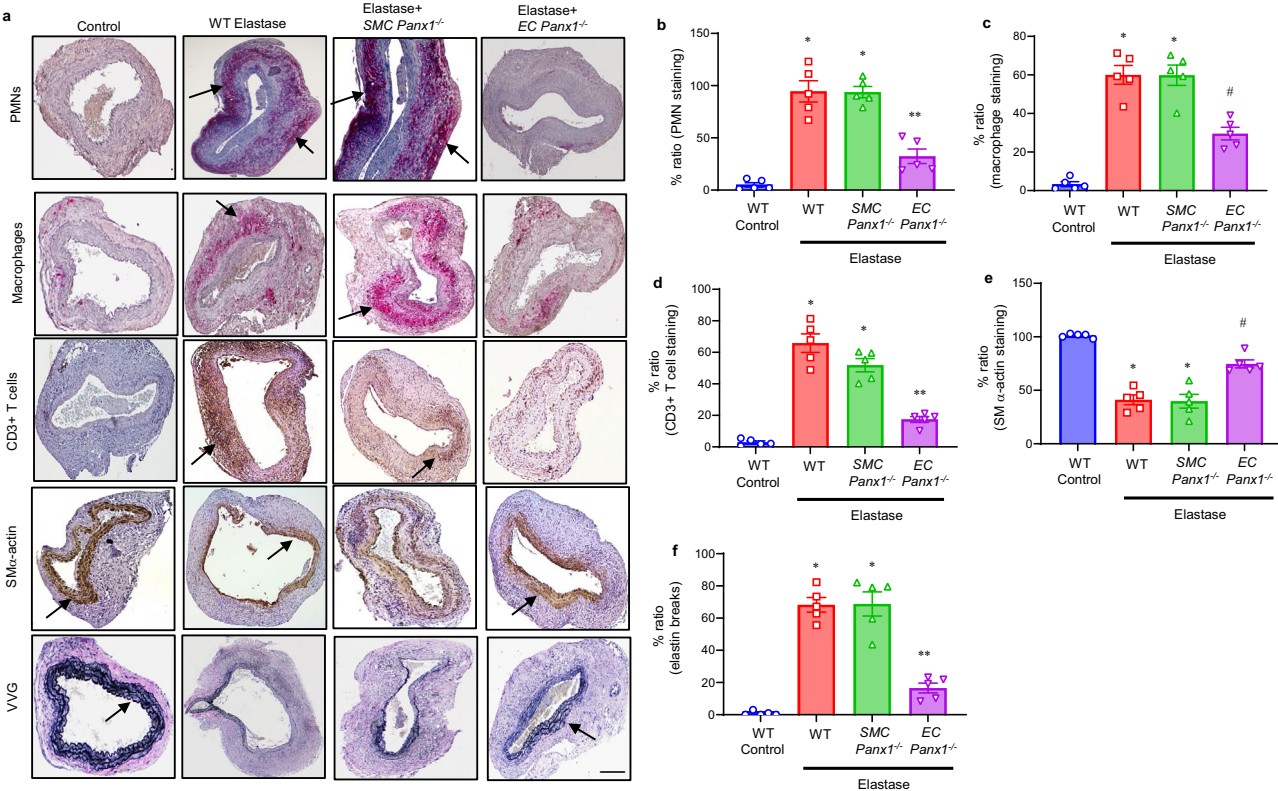

**Fig. 2 Panx1 deletion on endothelial cells mitigates leukocyte trafficking in AAA. a** Comparative histology and immunohistochemistry performed on day 14 indicates a marked decrease in CD3 + T cell, neutrophil (PMN), and macrophage (Mac-2) immunostaining, increase in smooth-muscle-cell α-actin (SMα-actin) expression, and decrease in elastic-fiber disruption in aortic tissue (Verhoeff–Van Gieson, VVG staining for elastin) of elastase-treated *EC-Panx1*$^{-/-}$ mice compared with elastase-treated WT and *SMC-Panx1*$^{-/-}$ mice. *n* = 5/group. Arrows indicate areas of immunostaining. Scale bar is 200 µm. **b–f** Quantification of immunohistochemical staining demonstrating a significant decrease in neutrophils (PMNs), macrophages, CD3 + T cells, and elastin degradation (VVG) staining, as well as increase in smooth muscle α-actin expression in elastase-treated *EC Panx1*$^{-/-}$ aortic tissue compared with elastase-treated WT and *SMC-Panx1*$^{-/-}$ mice. No significant differences were observed between elastase-treated WT and *SMC-Panx1*$^{-/-}$ mice. *P < 0.0001 vs. WT control; **P < 0.0001 vs. WT elastase and *SMC-Panx1*$^{-/-}$; #P = 0.0003 vs. WT elastase and *SMC-Panx1*$^{-/-}$. Data is represented as mean values ± SEM and comparative statistical analyses were done by one-way ANOVA.

PBN (500 μM) resulted in significant decrease in IL-β (3.0 ± 0.7 and 2.7 ± 0.4 pg/ml, *p* < 0.001) and HMGB1 (1.3 ± 0.2 and 1.5 ± 0.3 ng/ml, respectively) release from macrophages (Fig. 5f,g). Pretreatment of macrophages with P2X7 inhibitor (A80, 50 μM) before CMT from elastase-treated ECs resulted in a significant decrease in IL-1β (1.9 ± 0.4 pg/ml, *p* < 0.001) and HMGB1 (3.3 ± 0.6 ng/ml) secretion compared with CMT alone, respectively. No significant changes were observed in cytokine expression after macrophage treatments with P2Y2 inhibitor (Supplementary Fig. 4).

Previous studies indicate that mitochondrial (mt) DNA release in cytoplasm is associated with NLRP3 inflammasome activation[28]. Therefore, we hypothesized that EC Panx1-mediated ATP release can stimulate mtDNA release to mediate IL-1β and HMGB1 release by macrophages. CMT from elastase-treated ECs to macrophages was performed and then the cytoplasmic fraction was extracted, and mitochondrial DNA levels were assessed by qPCR with primers for cytochrome-c oxidase-1 gene. A significant increase in mtDNA release was observed in macrophages after CMT from elastase-treated ECs, which was inhibited by pretreatments of ECs with PBN or apyrase, as well as pretreatment of macrophages with P2X7 inhibitor (Fig. 5h). These results show that Panx1 stimulates ATP release from ECs that can activate P2X7 receptors on macrophages to initiate mtDNA release and IL-1β and HMGB1 secretion (Fig. 5i).

**EC-Panx1-mediated release of ATP modulates SMC activation and remodeling.** One feature of the pathogenesis of AAA formation is inflammation and vascular remodeling of SMCs. To investigate the effect of Panx1 on intracellular calcium responses in SMCs, we measured intracellular calcium influx in a time-dependent study. SMCs were separately treated with transient elastase exposure (0.4 U/ml) and recombinant ATP (1 μM) with/without pretreatment with PBN, GSK2 (a TRPV4 specific antagonist, 1 μM, one hour prior to exposure with elastase and ATP treatment), GSK1 (a TRPV4-specific agonist, 20 nM), or P2Y2 inhibitor (AR-C11, 50 μM), and intracellular Ca$^{2+}$ signal was analyzed. The elastase-induced increase in intracellular Ca$^{2+}$ of SMCs was significantly attenuated by TRPV4 or P2Y2 inhibition, but not by PBN treatment. Conversely, treatment with GSK1 (TRPV4 specific agonist) significantly increased Ca$^{2+}$ signal in the presence of elastase and ATP compared with elastase +ATP treatment alone (Fig. 6a). These results suggest that P2Y2 and TRPV4 inhibition can decrease elastase and purinergic signaling-induced stimulation of intracellular Ca$^{2+}$, while Panx1 inhibition of SMCs had no effect (Fig. 6a).

To delineate the EC–SMC crosstalk via Panx1/ATP signaling, CMT was performed from elastase-treated ECs (with/without apyrase or PBN pretreatments) to SMCs (with/without P2Y2 or GSK2 pretreatments) (Fig. 6b). CMT from elastase-treated ECs to SMCs induces a multifold increase in MCP-1, CXCL1, MIP-1α, RANTES, and IL-6, as well as MMP2 activity, which was

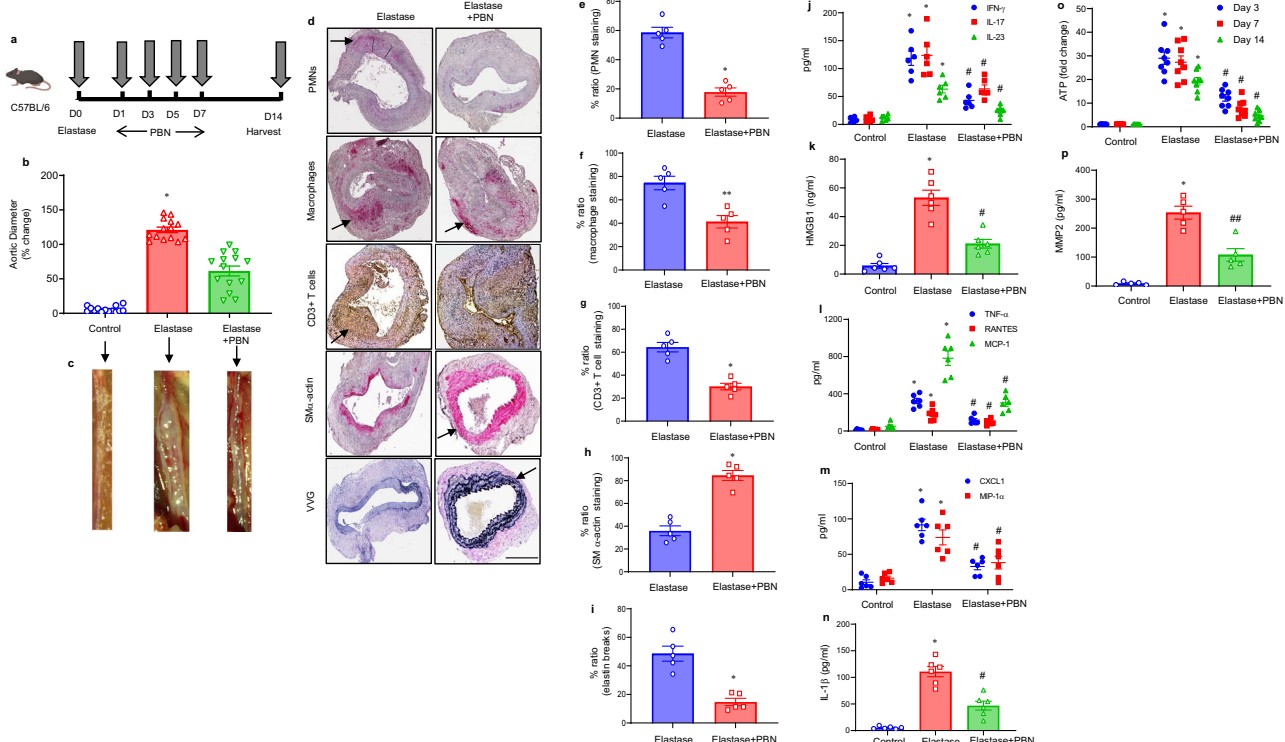

**Fig. 3 Pharmacological inhibition of Panx1 mitigates AAA formation in the elastase-treatment model. a** Schematic depicting treatment protocol for probenecid (PBN) in the elastase model of AAA. **b** PBN treatment attenuates aortic diameter in elastase-treated male WT mice compared with elastase treatment alone. *$P = 0.0001$ vs. all other groups; n = 14/group. **c** Representative images of aortic phenotype in the respective groups. **d–i** Comparative histology displayed a marked decrease in leukocyte infiltration and elastin-fiber disruption, as well as increase in SMα-actin expression in PBN-treated mice compared with elastase-treated mice alone.*$P = 0.007$; **$P = 0.01$; n = 5/group. Arrows indicate areas of immunostaining. Scale bar is 400 μm. **j–n** Pro-inflammatory cytokine expression in aortic tissue is significantly attenuated in elastase-treated WT mice after PBN administration compared with elastase-treated mice alone. *$P < 0.0001$ vs. respective controls; #$P < 0.001$ vs. elastase.; n = 6/group. **o** A significant decrease in aortic tissue ATP content was observed in PBN-treated mice compared with elastase-treated mice alone. *$P < 0.0001$ vs. controls; #$P < 0.0001$ vs. elastase; n = 8 mice/group. **p** Expression of MMP2 in aortic tissue was significantly attenuated in PBN-treated mice compared with elastase-treated mice alone. *$P < 0.0001$ vs. control; #$P = 0.0003$ vs. elastase; n = 5/group. Data are represented as mean values ± SEM and comparative statistical analyses were done by one-way ANOVA followed by multiple comparisons. Source data are provided as a Source Data file.

significantly attenuated by pretreatment of ECs with apyrase or PBN, as well as pretreatment of SMCs with P2Y2 or TRPV4 (GSK2) inhibitors (Fig. 6c–h). No significant changes were observed in cytokine expression or MMP2 activity after SMC treatments with inhibitors of P2X4, TRPV1, TRPC4, or TRPC6 (Supplementary Fig. 5 and 6). These results demonstrate that EC-induced SMC activation is mediated by EC-specific Panx1/ATP release that activates P2Y2 receptors on SMCs to increase intracellular Ca$^{2+}$ (Fig. 6i).

**Panx1 inhibitors are associated with decreased mortality in aortic aneurysm patients.** Aortic tissue from AAA patients demonstrated a multifold increase in mRNA expression of *PANX1* compared with controls (aortic tissue from organ-transplant donors) (Supplementary Fig. 7). A retrospective data analysis was performed on all adults with the diagnosis of aortic aneurysm (AA) using International Diagnosis Codes (ICD 9 or 10) between 1995 and 2015. Patients were identified using an institutional clinical data repository. Patients were stratified by use of medications known to affect pannexin channels, including probenecid and spironolactone[25,26,29]. The risk-adjusted long-term survival of AA patients with medication (probenecid or spironolactone) was significantly higher than the AA patients without these pannexin inhibitors ($p = 0.02$, Supplementary Fig. 8). These data suggest a possible association between Panx1

inhibition and decreased risk of mortality in aortic aneurysm patients.

## Discussion

The current study defines a mechanistic aspect of AAA formation mediated by Panx1 channels on endothelial cells as a pivotal regulator of aortic inflammation and vascular remodeling observed during the pathogenesis of aneurysm formation. Our results demonstrate that inducible, specific deletion of Panx1 from endothelial cells offers significant protection from AAA formation (i.e., decreased aortic diameter, pro-inflammatory cytokine expression, leukocyte infiltration, and MMP2 activity). Furthermore, pharmacological treatment with a Panx1 inhibitor (probenecid) substantially mitigates aortic inflammation and remodeling to attenuate aneurysm formation in two separate established models of murine AAA. In vitro experiments demonstrated the ability of aortic ECs to regulate macrophage activation via Panx1-dependent ATP secretion that upregulates P2X7Rs and mtDNA release to increase the secretion of IL-1β and HMGB1. Also, Panx1/ATP signaling initiated by ECs modulates SMC activation via P2Y2Rs to upregulate cytokine secretion as well as matrix metalloproteinase activity that is mediated via TRPV4 channels. These findings were supported by in vivo measurements in which elevated ATP levels, macrophage-specific cytokine secretion, and SMC-specific MMP2 activities were

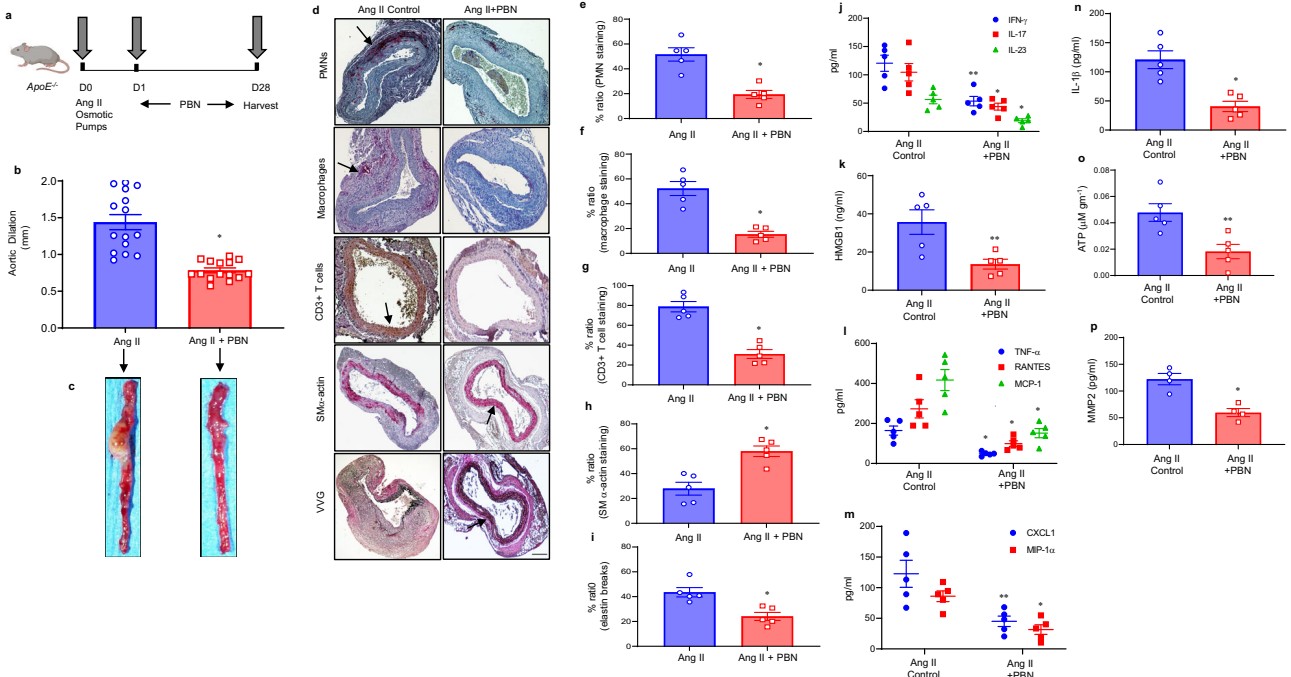

**Fig. 4 Inhibition of Panx1 channels attenuates AAA formation in the Ang-II model. a** Schematic depicting the Ang-II model of AAA in *ApoE*−/− mice. Osmotic pumps with either Ang II or saline (control) were inserted into the subcutaneous tissue of mice with/without treatment with PBN. Aortic diameter was measured on day 28, and tissue was harvested for further analysis. **b** Ang-II mice treated with PBN demonstrated significantly decreased aortic diameter compared with angiotensin treated mice alone. *P = 0.001; n = 15 mice/group. **c** Representative images of aortic phenotype in all groups. **d–i** Comparative histology performed on day 28 demonstrates that Ang-II-treated *ApoE*−/− mice administered with PBN have decreased polymorphonuclear neutrophil (PMNs), macrophage (Mac-2), CD3 + T-cell infiltration, and elastic-fiber disruption (Verhoeff–Van Gieson staining), and increase in SM-α-actin expression compared with mice treated with Ang II alone. Arrows indicate areas of immunostaining. *P = 0.007; n = 5 mice/group. Scale bar is 200 μm. **j–n** Aortic inflammation is mitigated by Panx1 antagonism in the Ang II model of AAA. Aortic tissue from Ang-II-treated *ApoE*−/− mice after PBN administration showed a significant attenuation in pro-inflammatory cytokine/chemokine production compared with Ang-II-treated mice alone. *P = 0.007 vs. respective Ang-II controls; **P = 0.01; n = 5 mice/group. **o** A significant decrease in aortic tissue ATP content was observed in PBN-treated mice compared with Ang-II-treated mice alone. *P = 0.01; n = 5 mice/group. **p** Expression of MMP2 in aortic tissue was significantly attenuated in PBN-treated mice compared with elastase-treated mice alone. *P = 0.02; n = 4/group. Data are represented as mean values ± SEM and comparative statistical analyses were done by two-tailed *t*-test.

attenuated in WT mice by Panx1 antagonism, as well as in tamoxifen-treated *EC-Panx1*−/− mice. Finally, the clinical relevance of these findings is enhanced by increased PANX1 expression observed in AAA patients, as well as a correlative retrospective study showing a significant association with overall decreased mortality in AA patients with Panx1 inhibitors.

Previous studies from our group and others have elucidated that tissue inflammation plays a key role in the pathogenesis of AAA[5,20–22,27,30,31]. The chronic inflammatory process is characterized by unregulated matrix-metalloproteinase activity, increased cytokine production, elastin degradation, as well as smooth-muscle apoptosis[32,33]. The collective effect of these multifaceted pathways leads to progressive weakening of the aortic vascular wall, dilation of the vessel, and eventual aortic rupture that can lead to sudden death[34,35]. The UK Small Aneurysm trial has shown that the duration of clinical observation to surgery is generally around 5 years, that represents leaving a crucial period for possible therapeutic intervention to mitigate AAA expansion and impending rupture to prevent mortality[36–40]. This initial period of diagnosis represents a potential duration for pharmacologic intervention to modulate the pathophysiology of aortic remodeling via ion-channel regulation of vasculature, i.e., panx1 antagonists, to significantly decrease the rate of aortic expansion.

Recent studies demonstrated that ATP can be released in a controlled manner through hemichannels such as pannexins. The primary functions of pannexin channels are postulated to be the release of purines from intracellular stores. The Panx1 isoform is expressed on endothelial cells as well as in smooth-muscle cells of arteries[41]. Previous studies have demonstrated that ATP release by Panx1 channels regulates vascular function in venous and arterial vessels[14,42]. Endothelial Panx1 can regulate TNF-α-induced leukocyte adhesion and emigration in venules, as well as modulate pulmonary and cerebral ischemia/reperfusion injury[14,18,43]. Therefore, we specifically delineated the contributions of EC– and SMC–Panx1 on aortic inflammation and vascular remodeling in AAA formation. Our studies demonstrate for the first time that deletion of EC-specific Panx1, but not SMCs, significantly reduces the aortic diameter in a murine AAA model. These protective effects appear to be mediated through multiple, seemingly interconnected signaling pathways involving immune cells, i.e., macrophages, neutrophils, and T cells, as well as aortic SMCs. We have previously shown AAA formation to be regulated via macrophage activation, specifically inflammasome activation via IL-1β as well as increased HMGB1 secretion, both of which were significantly decreased in *EC-Panx1*−/− mice[20–22]. Similarly, AAA formation involves SMC activation, matrix-metalloproteinase activity, and apoptosis, which were downregulated in *EC-Panx1*−/− mice[44–46].

Panx1 can be a conduit for multiple molecules across the plasma membrane, i.e., ATP and UTP, among other secretomes[13]. Purines (e.g., ATP, ADP, and adenosine) can exert

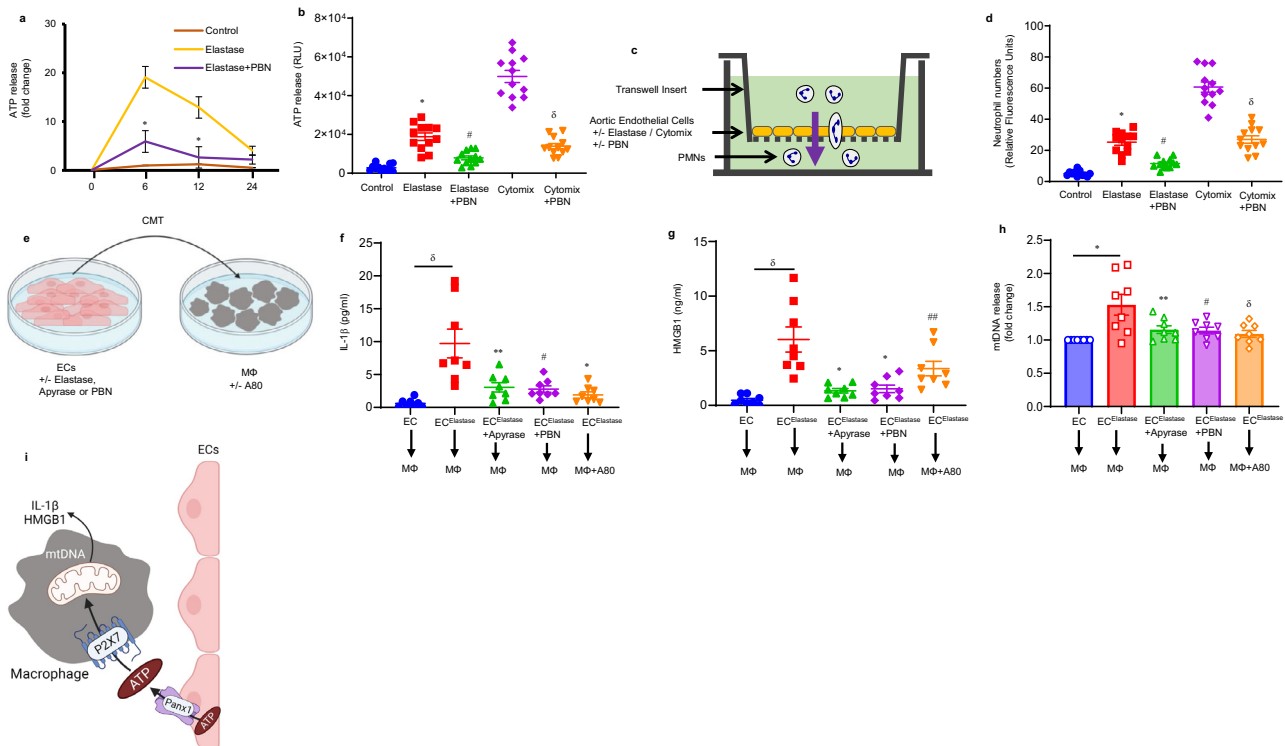

**Fig. 5 Panx1-dependent ATP release from ECs modulates neutrophil transmigration and macrophage activation. a** Transient elastase-treatment-induced eATP release from EC cultures is significantly mitigated by PBN treatment at 6- and 12 h. *$P < 0.0001$ vs. elastase; $n = 12$/group. **b** Elastase- and/or cytomix-induced eATP release by ECs is attenuated by Panx1 inhibition via PBN treatment compared with untreated controls. * $P < 0.0001$ vs. control; #$P = 0.0009$ vs. elastase; δ$P < 0.0001$ vs. cytomix; $n = 12$/group. **c** Schematic showing the in vitro transwell model to demonstrate transendothelial migration of polymorphonuclear neutrophils (PMNs). Fluorescent-labeled primary murine-derived PMNs were included in the top chamber containing ECs and cocultured for 24 h. **d** Exposure to elastase or cytomix showed a significant increase in neutrophil transmigration, which was significantly attenuated by pretreatment with PBN. *$P < 0.0001$ vs. control; #$P = 0.0002$ vs. elastase; δ$P < 0.0001$ vs. cytomix; $n = 12$/group. **e** Schematic depicting conditioned media transfer (CMT)-based experiments from elastase-treated ECs to macrophages with/without pretreatment with inhibitors. **f**, **g** CMT from elastase-treated ECs to macrophages induces a significant upregulation of IL-1β and HMGB1 secretion, which was blocked by pretreatment of ECs with apyrase or PBN, as well as pretreatment of macrophages with P2X7 inhibitor (A80). δ$P < 0.0001$ vs. EC → MΦ; *$P = 0.0001$, **$P = 0.0009$, #$P = 0.0005$ and ##$P = 0.03$ vs. EC^elastase → MΦ; $n = 8$/group. All comparative data above are represented as mean values ± SEM and statistical analyses were done by one-way ANOVA. **h** Panx1-dependent ATP release from ECs stimulates mtDNA release that is prevented by PBN and P2X7R inhibition. CMT from elastase-induced ECs was performed on macrophages. DNA was isolated from cytosolic fractions of macrophages and the levels of mitochondrial (mt)DNA were analyzed by quantitative RT-PCR. CMT from elastase-treated ECs to macrophages induces a significant upregulation of mtDNA release, which was blocked by pretreatment of ECs with apyrase or PBN, as well as pretreatment of macrophages with P2X7 inhibitor. *$P = 0.0005$; **$P = 0.02$, #$P = 0.01$, and δ$P = 0.004$ vs. EC^elastase → MΦ; $n = 8$/group. Data are presented as fold change relative to EC → MΦ and compared using two-tailed Wilcoxon test. **i** Schematic displaying the crosstalk between ECs and macrophages via Panx1-dependent release from elastase-induced ECs that stimulates P2X7R and mtDNA release from macrophages.

an important role as extracellular signaling molecules mediated via GPCR or ligand-gated ion-channel purinergic receptors[47,48]. ATP extracellularly can act as a chemotactic "danger" signal (i.e., DAMP molecule) for leukocytes and signal via binding to purinergic P2 (P2X or P2Y) receptors. The cellular response to inciting stimuli such as shear stress or inflammation depends upon the interplay of purinergic P2X (ATP) and P2Y (ATP, ADP, and UTP) receptor activation, as well as P1 (adenosine) receptors in specific tissue. The ratio of these signaling molecules and the relative expression of purinergic receptors can mediate pathological conditions triggering inflammation[49]. Activation of P2X7 receptors in response to ATP has been shown to upregulate mitochondrial DNA release and IL-1β as well as HMGB1 secretion[28]. Our studies exhibited this phenomenon to be regulated by crosstalk with EC Panx1-mediated ATP release that can be blocked by probenecid treatment. Second, P2Y2 is known to be highly expressed in SMCs and ATP has been shown to stimulate aortic SMCs predominantly at the P2Y2 receptor via JNK-signaling pathway[50–52]. The conceptual framework of this

study deciphers how Panx1 on endothelial cells is involved in initiation and propagation of intercellular calcium waves in the neighboring aortic SMCs. Mechanical stress (or other stimuli, like depolarization) during vascular remodeling of aortic tissue opens Panx1 channels, enabling the exit of cytoplasmic ATP. Extracellular ATP activates purinergic P2Y2 receptors on cells within diffusion distance, i.e., SMCs. P2Y2 receptors are G-protein-coupled and activate phospholipase C, resulting in IP3 release and increased calcium from intracellular stores[53] but also are free to permeate gap-junction channels to contiguous cells. The transient increase in cytoplasmic calcium concentration following P2Y2 stimulation can sensitize TRPV4 channels, resulting in an increase in influx of extracellular calcium. Therefore, we postulate that destabilization of intracellular $Ca^{2+}$ homeostasis facilitates increased MMP2 activity[46,54] and this cascade leads to SMC degradation and subsequent vascular remodeling resulting in AAA formation.

Several pharmacological tools have been used in studies of pannexin-channel physiology. Probenecid (an organic anion

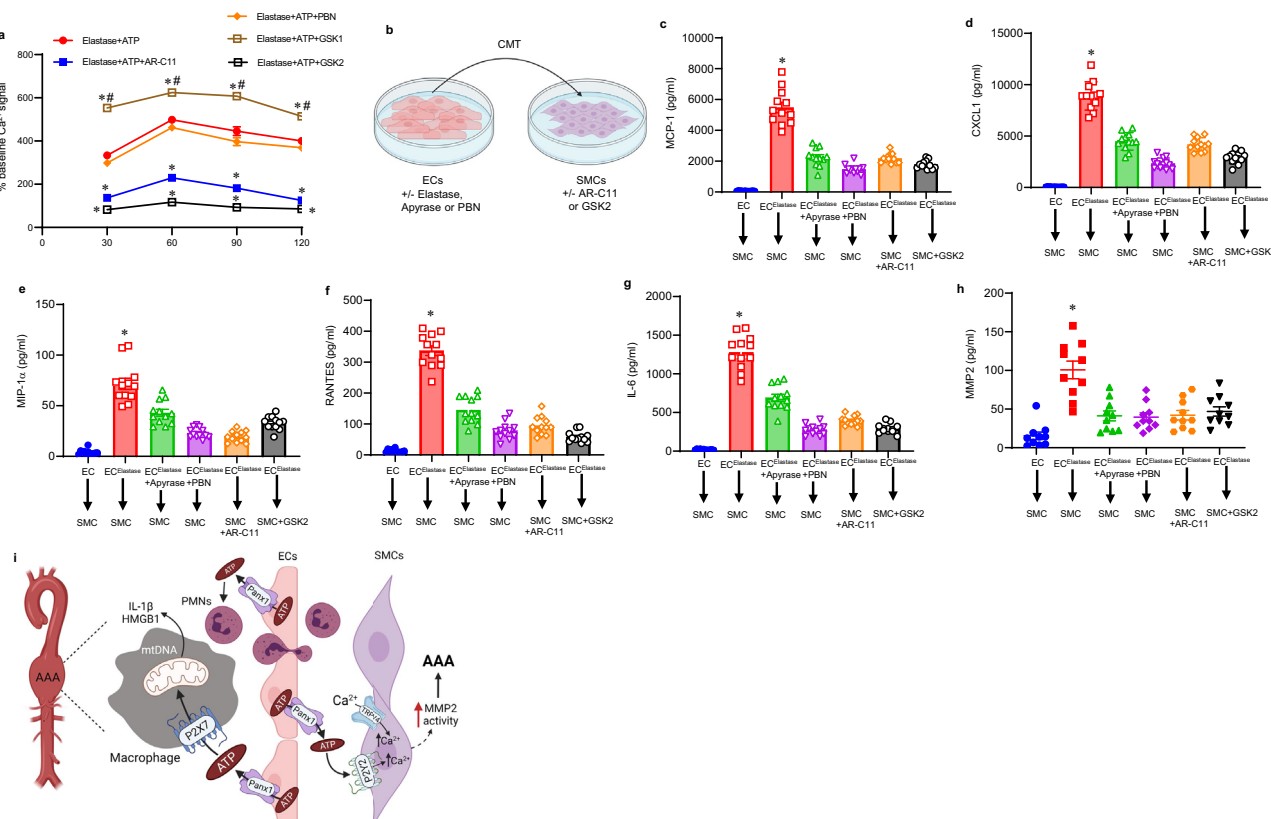

**Fig. 6 Panx1-dependent ATP release from ECs results in SMC activation that is inhibited by P2Y2 and TRPV4 antagonism. a** An increase in Ca$^{2+}$-influx was observed in elastase and recombinant ATP- (1 μM) treated SMCs that was inhibited by P2Y2 (AR-C11) and TRPV4 antagonist (GSK2) blockade, but increased after TRPV4 agonism by GSK1 treatment. The Ca$^{2+}$ signal after treatment was expressed as % change intensity compared with baseline in untreated cells. *$P < 0.0001$ vs. elastase+ATP; #$P < 0.0001$ vs. elastase+ATP + GSK2; $n = 12$/group. **b** Schematic depicting conditioned media-transfer (CMT)-based experiments from elastase-treated ECs to SMCs with/without pretreatment with inhibitors. **c**–**g** CMT from elastase-treated ECs to primary SMCs induces a significant upregulation of MCP-1, CXCL1, MIP-1α, RANTES, and IL-6 secretion, which was blocked by pretreatment of ECs with apyrase or PBN, as well as pretreatment of SMCs with P2Y2 or TRPV4 inhibitors, respectively. *$P < 0.0001$ vs. all other groups; $n = 12$/group. **h** Primary SMCs were treated with CMT from elastase-treated ECs and MMP2 activity secretion was analyzed in cell culture supernatants after 24 h. A multifold increase in MMP2 activity was observed in SMCs after CMT from elastase-treated ECs compared with controls, and was significantly attenuated by pretreatment of ECs with apyrase or PBN, as well as pretreatment of SMCs with P2Y2 or TRPV4 inhibitors, respectively. *$P < 0.0001$ vs. other groups; $n = 8$/group. **i** Schematic representation of signaling events mediated by Panx1 signaling during the pathogenesis of AAA. ATP release from Panx1 channels in aortic ECs modulates macrophage activation via P2X7 receptors, and stimulates mtDNA release, leading to IL-1β and HMGB1 secretion thereby causing aortic inflammation. Also, Panx1-mediated ATP release from ECs activates P2Y2 receptors and TRPV4 channels on SMCs to stimulate pro-inflammatory cytokine secretion and destabilization of intracellular Ca$^{2+}$ homeostasis that facilitates increased MMP2 activity resulting in aortic remodeling. Panx1/ATP signaling also contributes to leukocyte trafficking in the aortic wall, as observed by transendothelial migration of neutrophils. HMGB1, high-mobility-group box1; PMNs, neutrophils; ECs, endothelial cells; SMCs, smooth-muscle cells; mtDNA, mitochondrial DNA. Comparative data are represented as mean values ± SEM and statistical analyses were done by one-way ANOVA.

transport inhibitor) exhibits much greater selectivity against pannexin currents than against connexin currents (a very closely related conductance)[55]. Although probenecid has been traditionally used in the treatment of gout, but recent studies suggest that it can alleviate hypertension via inhibition of α-adrenergic receptor-mediated vasoconstriction, thereby contributing to the regulation of peripheral vascular resistance[29,56]. Furthermore, spironolactone has been shown to be an effective antihypertensive drug and can robustly inhibit both mouse and human Panx1 channels[26]. Our clinical data indicate a broad correlation between the use of these Panx1 inhibitors and decrease in overall mortality in AAA patients. Previous reports from our group and others suggest that pannexin-1/ATP-signaling pathway can participate in the regulation of vascular tone and blood pressure[57–60]. Therefore, the reduction of arterial pressure is a Panx1-dependent action that is plausibly associated with the decrease in aneurysm formation and subsequent risk of death from aneurysm rupture.

While our clinical meta-analysis does not decipher between the direct versus indirect effects of Panx1-specific inhibitors in reducing mortality of AA patients, but it provides associative evidence of a therapeutic option by the use of repurposed medications like probenecid and spironolactone in treating these patients. The clinical relevance of these inhibitors via mechanosensitive modulation of Panx1 channels in aortic tissue and/or by regulation of reduction of arterial pressure remains to be deciphered in the patient population.

Our recent study demonstrated that pharmacological inhibition of TRPV4 channels can regulate endothelial cell-dependent permeability as well as smooth-muscle-cell-dependent pro-inflammatory cytokine secretion[46]. The present study demonstrates an interesting correlation between Panx1/ATP signaling from ECs to modulate SMC activation via TRPV4 channels. Collectively, these data point to a cascade of ionic conductances mediated by an interplay between Panx1, TRPV4, and ATP-

ligand gated channels, leading to elevation of $Ca^{2+}$ and ATP release. The modulation of electrophysiological responses in the dynamic crosstalk between ECs and SMCs via pharmacological compounds (probenecid, spironolactone, and/or GSK2) represents a developing strategy to counter the wide variety of stimuli (membrane stretch/strain, membrane depolarization) with consequent activation of mechanosensitive ion channels and purinergic P2X/Y receptors, in chronic inflammatory-disease processes.

Collectively, these data identify a mechanistic role of Panx1 channels in aortic aneurysm formation. These data have implications for how changes in ion channels modify their extracellular environment through the selective release of molecules that can exert multifaceted effects to trigger tissue inflammation and remodeling. The mechanistic signaling involving the concurrent activation of Panx1, P2X/Y, and TRPV4 channels could account for the slow and sustained development of chronic aortic pathologies, i.e., aortic aneurysms. The ability of targeted inhibitors to immunomodulate this complex electrophysiological response may represent repurposing of existing therapeutic agents that may have clinical implications in vascular diseases.

## Methods

**Study design and approval.** Collection of human aortic tissue was approved by the Institutional Review Board, University of Florida (protocol # 13178). Preoperative consent was obtained from all patients. AAA tissue from male patients was resected during open surgical AAA repair, and abdominal aortic tissue was obtained from transplant-donor patients to serve as controls. For the clinical data retrospective analyses, the University of Virginia institutional review board approved this study (protocol #17900). All adults with the diagnosis of aortic aneurysm using International Diagnosis Codes (ICD 9 or 10) between 1995 and 2015 were reviewed. All animal experiments were conducted according to the National Institutes of Health guidelines and were conducted under animal protocols approved by the University of Florida's Institutional Animal Care and Use Committee (protocol #202110051). All experiments started from a testable hypothesis or prediction and experiments were planned without bias on the outcome and performed in a blinded manner. Mice were electronically tagged to facilitate with the blinding process. To ensure randomization, we used an I.D. code in conjunction with our in-house scripts to assign animals prior to surgery to treatment groups, and this number was used for all samples generated from these animals, which were unblinded at the end of each experimental series. Mice were housed in a pathogen-free animal facility under a 12 h light/dark cycle at constant temperature and humidity, and fed standard rodent chow and water ad libitum. Outcome assessment and scoring of animals for all samples was performed using automated software or by blinded observers. For microscopy, we used independent observers using random field selection and image analysis.

**Human AAA tissue and clinical data analysis.** Aortic tissue from AAA patients and control (organ-transplant donors) was homogenized, and mRNA extraction was performed using a RNeasy Mini Kit (Qiagen, Valencia, CA). mRNA was reverse-transcribed to cDNA using the procedures and reagents provided in the iScript cDNA Synthesis Kit (Bio-Rad). GAPDH was used as the positive control in conjunction with the *PANX1* primers (Thermo Fisher Scientific). Primers used for quantitative RT-PCR for human *PANX1* were CCACGGAGTACGTGTTCTCG (RT-F) and CCGCCCAGCAATATGAATCC (RT-R). qRT-PCR was performed and analyzed using a CFX Connect Real-Time PCR Detection System (Bio-Rad CFX96 Real-Time System). Cycle-threshold data from the CFX system were analyzed and reported using the $2^{\Delta Ct}$ relative to GAPDH quantification method. For clinical data analysis, aortic aneurysm patients were identified using an institutional Clinical Data Repository (CDR) capturing all patient visits at University of Virginia Health System. This database included demographic data as well as medical comorbidities, prescription drug information, operative records, and long-term survival captured through the Virginia Health Department using social-security records. Patients were stratified by use of medications demonstrated to affect pannexin channels, including probenecid and spironolactone.

**Animals and reagents.** This study used 8–12-wk-old male C57BL/6 WT mice (Jackson Laboratory, Bar Harbor, ME). The inducible, endothelial-specific Panx1-knockout mice, $VE\text{-}Cad\text{-}CreER^{T2+}Panx1^{fl/fl}$ mice ($EC\text{-}Panx1^{-/-}$), were generated by crossing vascular endothelial *(VE)-Cad-Cre*$^{ERT2+}$ mice with $Panx1^{fl/fl}$ mice, as previously described and characterized[14]. The inducible smooth-muscle-cell-specific Panx1-knockout, $SMMHC\text{-}CreER^{T2}/Panx1^{fl/fl}$ ($SMC\text{-}Panx1^{-/-}$) mice were generated using $SM\text{-}MHC\text{-}Cre^{ERT2+}$ mice with $Panx1^{fl/fl}$ mice, as previously described[42]. To induce *Panx1* deletion in the cell-specific knockout mice, intraperitoneal injections of tamoxifen (1 mg in 0.1 ml of peanut oil) were administered

for 10 consecutive days before the experiment. *EC-Panx1*$^{-/-}$ mice and *SMC-Panx1*$^{-/-}$ injected with peanut oil (vehicle for tamoxifen) served as littermate controls. WT mice were also treated with Panx1 inhibitor, probenecid (1.1 mg/kg intraperitoneally) on days 1, 3, 5, and 7. A murine elastase-treatment model of AAA formation was used, as previously described[61]. The abdominal aorta was treated topically with 30 μl of type-1 porcine pancreatic elastase (5 U/mg of protein) on day 0. On day 14 following elastase treatment, the abdominal aorta was measured by video micrometry and expressed as percentage increase over baseline aortic diameter. Aortic-diameters were measured by video micrometry using NIS-Elements D5.10.01 software attached to the microscope (Nikon SMZ-25 with NIS elements D5.10.01; Nikon Instruments, Melville, NY). Aortic dilation percentage was determined by [(maximal AAA diameter – self-control aortic diameter)/(self-control aortic diameter)] × 100. Aortic dilation of ≥100% was considered positive for AAA.

**Angiotensin-II model of AAA formation.** Osmotic pumps (Alzet 2004; Durect, Cupertino, CA, USA) containing angiotensin II (AngII, 1000 ng/kg/min, Sigma-Aldrich) were placed subcutaneously into 8–12-week-old *ApoE*$^{-/-}$ male mice, as previously described[46,62]. All mice were fed unlimited water and placed on a high-fat diet (TD 88137, Harlan Teklad, Indianapolis, IN, USA) with no restrictions on movement. A separate group of mice were treated with intraperitoneal injections of PBN from day 1 to day 27. Aortic diameters in all groups were measured and aortic tissue was harvested on day 28 for further analyses.

**Histology.** Murine abdominal aortas were procured and placed in 4% paraformaldehyde for 24 h, and embedded in paraffin, and sections were stained by immunohistochemistry as previously reported[63]. Antibodies for immunohistochemical staining were anti-mouse Mac2 for macrophages (1:10,000, Cedarlane Laboratories, Burlington, ON, Canada), CD3 for T cells (1:500, Santa Cruz Biotech, Dallas, TX), anti-mouse neutrophils for polymorphonuclear neutrophils (PMNs) (1:10,000, AbD Serotec, Oxford, United Kingdom), and anti-mouse α-smooth-muscle-actin (α-SMA, 1:1000, Sigma, St. Louis, MO). Aortic sections were also stained with hematoxylin and eosin, and Verhoeff-Van Gieson for elastin (Polysciences, Inc., Warrington, PA). Visualization color development was completed using diaminobenzidine (Dako, Glostrup, Denmark). Images were acquired using AxioCam Software version 4.6 and an AxioCam MRc camera (Carl Zeiss Inc., Thornwood, New York). Threshold-gated positive signal was detected within the AOI and quantified using Image-Pro Plus version 7.0 (Media Cybernetics Inc., Bethesda, MD).

**Immunofluorescent staining for Panx1.** Fixed aortic tissue was paraffin-embedded, and 5-μm sections were obtained. Serial tissue sections were subjected to paraffin removal followed by antigen retrieval (Antigen Unmasking Solution, Vector) and 1 h blocking in 5% normal goat serum, 0.2% Triton X-100, and 0.05% fish-skin gelatin. Aortic sections were stained with Panx1 antibody (1:50, Sigma Aldrich, St. Louis, MO), or secondary antibody only as previously described by our laboratory[43]. Anti-rabbit secondary F(ab)′2 antibody coupled to Alexa Fluor 568 (1:400, Life Technologies, Carlsbad, CA) was applied to all sections followed by the addition of DAPI ProLong Gold Antifade reagent (Invitrogen, Carlsbad, CA) and imaged with an Olympus FluoView 1000 (Olympus America, Center Valley, PA) as previously described[64].

**ATP measurements.** ATP levels were measured in cell culture supernatants, plasma, or aortic tissue using a luciferase-based ATP bioluminescence assay kit, per the manufacturer's recommendations (Sigma-Aldrich, St. Louis, MO). For extracellular ATP analysis in murine plasma, whole-blood sample of 500 μl was drawn from the infrahepatic inferior vena cava just lateral to the abdominal aorta into a syringe with EDTA. Plasma supernatants were collected after centrifugation for subsequent measurement of ATP content using the bioluminescence assay kit, as previously described[65]. Analysis was performed by calculation based on the standard curve, background subtraction, and normalization to controls.

**Endothelial cell culture and neutrophil-migration assay.** C57BL/6 murine primary aortic endothelial cells (ECs, Cell Biologics, Chicago, IL) were cultured overnight in endothelial cell-growth medium (Cell Biologics). ECs were grown in 6-well culture plates, and grown to 80–90% confluency. On the day of the experiment, cells were rinsed, then incubated in fresh culture media with the ectonucleotidase triphosphate diphosphohydrolase-1 (CD39/NTPDase 1) inhibitor, ARL67156 (300 μM, Tocris Bioscience) for 30 min at 37 °C. Cells were exposed to either transient elastase treatment for 5 min followed by washing the cells with PBS and replacing the media, or treated with cytomix (IL-1β + HMGB1 + IL-17, R&D Systems, Minneapolis, MN; 50 μM). Cell culture supernatants were collected and analyzed for eATP release as described above. A transmigration assay (Cell Biolabs, San Diego, CA) was used per the manufacturer's instructions to measure transmigration of neutrophils across the endothelium. Neutrophils were isolated from mouse spleens using a mouse Neutrophil Isolation Kit (Miltenyi Biotec, Auburn, CA), labeled with fluorescent LeukoTracker dye, and incubated with ECs treated with/without cytomix or PBN (Sigma Aldrich, St. Louis, MO, 500 μM). Conditioned media transfer (CMT) was performed using ECs and primary F4/80$^+$

macrophages (Miltenyi Biotec, Germany). ECs were grown to confluency in 6-well plates and exposed to transient elastase treatment with/without apyrase (10 U/ml, Sigma-Aldrich) or PBN (500 μM, Sigma-Aldrich). After 6 h, CMT was performed to macrophage cultures (treated with/without P2X7 inhibitor, A804598 (A80), 50 μM, Tocris) and culture supernatants were harvested after 24 h for analysis of IL-1β (R&D Systems) and HMGB1 expression (IBL International, Hamburg, Germany) by ELISA.

**Cytokine measurements**. Cytokine expression in murine aortic tissue homogenates and cell culture supernatants was quantified using the Luminex Bead Array technique using a multiplex cytokine-panel assay (Bio-Rad Laboratories, Hercules, CA).

**Isolation of primary aortic smooth-muscle cells and in vitro experiments**. Primary aortic smooth-muscle cells (SMCs) were purified from C57BL/6 mice as previously described[66]. SMCs were exposed to transient elastase treatment for 5 min followed by washing the cells with PBS and replacing the media with/without ATP (1 μM, Thermo Fisher Scientific, Waltham, MA), PBN (500 μM, Sigma-Aldrich), P2Y2 inhibitor (AR-C118925XX (AR-C11); 10 μM, Tocris Bioscience, Minneapolis, MN), GSK1016790A (GSK1, 20 nM, Tocris Bioscience), or GSK2193874 (GSK2, 1 μM, Tocris Bioscience) treatment. After 24 h, cell culture supernatants were collected and analyzed. After 24 h, cell culture supernatants were collected and analyzed for cytokine expression using a multiplex assay. Measurement of calcium concentration was performed using the Calcium 6 Assay Kit (Molecular Devices, San Jose, CA). Briefly, SMCs were plated with a density of $5 \times 10^5$ cells/well in RPMI-1640 medium with 10% heat-inactivated fetal bovine serum (Gibco, Grand Island, NY). Cells were transiently treated with elastase for 5 min followed by washing with phosphate-buffer saline (PBS) and replacement of the medium with/without PBN, 5-BDBD, GSK2, or A804598 (A80) treatment, and changes in intracellular fluorescence were measured using a microplate reader with excitation/emission wavelength (485/525 nm, BioRad microplate reader with Microplate Manager v6.3). Data are presented as the % change to baseline $Ca^{2+}$ signal measured every 30 s for a total of 2 min.

Conditioned media transfer (CMT) was performed using ECs and SMCs. ECs were grown to confluency in 6-well plates and exposed to transient elastase treatment with/without apyrase (10 U/ml, Sigma-Aldrich) or PBN (500 μM; Sigma Aldrich). After 6 h, CMT was performed to SMC cultures treated with/without P2Y2 (AR-C11, 10 μM), P2X4 (50 μM) or P2X7 (50 μM), TRPV4 (GSK2), TRPV1 (A425619), TRPC4 (ML204), or TRPC6 (SAR7334) at a dose of 1 μM each (all from Tocris Bioscience), and culture supernatants were harvested after 24 h for analysis of cytokines (MCP-1, CXCL1, MIP-1α, RANTES, and IL-6) and MMP2 activity (Luminex bead array, Millipore Sigma, St. Louis, MO).

**Mitochondrial DNA-release assay**. The mtDNA levels in the cytosol were measured as described previously[28]. Briefly, $1 \times 10^7$ peritoneal macrophages were homogenized with a homogenizer in the presence of protease inhibitor and then were subjected to centrifugation at 700 g for 10 min at 4 °C. Protein concentration and volume of the supernatant were normalized, followed by centrifugation at 10,000 g for 30 min at 4 °C for the production of a supernatant corresponding to the cytosolic fraction. DNA was isolated from 200 μL of the cytosolic fraction by using a DNeasy Blood & Tissue kit (Qiagen). The levels of mtDNA encoding cytochrome-c oxidase 1 were measured in triplicate experiments by quantitative real-time PCR with the same volume of the DNA solution (BioRad C1000 and CFX96 systems). The following primers were used: mouse cytochrome-c oxidase I: forward, 5′-GCCCCAGATATAGCATTCCC-3′, and reverse, 5′-GTTCATCCTGTTCCTGCTCC-3′.

**Statistics and reproducibility**. For the animal, cell culture, and human tissue studies, values are presented as the mean ± standard error of the mean (SEM), and statistical evaluation was performed using GraphPad Prism 6 software. The experiments were usually conducted at least three times independently to ensure reproducibility of the data and for statistical analysis, and the biological experiments were performed in independent replicates. One-way analysis of variance (ANOVA) after post hoc Tukey's test was used to determine the differences among multiple comparative groups. Unpaired t-test with nonparametric Mann–Whitney or Wilcoxon rank-sum test was also used for pairwise comparisons of groups. P-value less than 0.05 was considered statistically significant. For the retrospective human clinical data analyses, Mann–Whitney U test was used for independent continuous variables. Chi-square test was used for independent categorical variables. Kaplan–Meier survival analysis and Cox proportional hazards model was utilized to compare unadjusted and risk-adjusted long-term survival, respectively. A priori, the decision was made to include all variables captured by the CDR and no selection methods were used. Patients began accruing time in the database at the first diagnosis of aortic aneurysms. Statistical significance was determined by two-sided α of 0.05. All analyses were performed using SAS version 9.4 (SAS Institute, Cary, NC).

**Reporting summary**. Further information on research design is available in the Nature Research Reporting Summary linked to this article.

**Data availability**
All data supporting the findings of this study are available within the article, Supplementary Information or Source Data file. Source data are provided with this paper.

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

## Acknowledgements
This study was supported by research grant from National Institute of Health (NIH RO1 HL138931, GRU, and AKS; NIH PO1 HL120840, KSR, and BEI). Schematic images were created using biorender.com.

## Author contributions
A.C.F. and M.D.S. contributed equally to the work. A.C.F., M.D.S., C.T.E., G.S., V.L., Z.L., G.L., M.D.S., R.B.H., B.E.I., and A.K.S. conducted experiments and acquired data. J.H.M. acquired and analyzed the human clinical data. K.S.R. and B.E.I. provided transgenic mice. K.S.R., B.E.I., G.R.U., and A.K.S. edited the paper. G.R.U. and A.K.S. supervised experiments, analyzed the interpreted data, drafted the paper, designed the study, and coordinated integration of collaboration between all participating laboratories. All authors critically reviewed and edited the final version of the paper.

## Competing interests
The authors declare no competing interests.
