## [Peer Review File · Nature Communications]

Endothelial pannexin-1 channels modulate macrophage and smooth muscle cell activation in abdominal aortic aneurysm formationEditorial Note: Parts of this Peer Review File have been redacted as indicated to remove third-party material where no permission to publish could be obtained.

Reviewers' comments:

Reviewer #1 (Remarks to the Author):

The manuscript by Filiberto and co-authors investigates the role of endothelial Panx1 in aortic inflammation and remodeling during abdominal aortic aneurysms development. The high mortality rate among patients with AAA represents an urgent and unaddressed clinical problem with a low success rate of surgical intervention as the only treatment option. An urgent need for new preventive therapies targeting inflammatory and proteolytic pathways in the aorta underscores the clinical relevance of this research.

The authors use two animal models of AAA and perform in vitro experiments to define the role of Panx1 in AAA pathology, and define its downstream partners in ECs and SMCs that are critical players for neutrophil transmigration, macrophage activation, SMC activation and remodeling. The study tests a hypothesis that Panx1 plays a key mechanistic role in the regulation of these pathological pathways and can serve as a target to alleviate AAA progression. Observations performed in vivo are dissected down to distinct cell-type-specific interactions. Another line of evidence supporting their hypothesis is produced by retrospective meta-analysis of clinical data in human patients with AAA who received off-label treatments with two drugs that have Panx1-inhibitory activity.

Overall, this is a well-designed and written manuscript. However, several major and minor concerns dealing with data potential indirect effects of elastase on Panx1 activity proposed mechanisms of NLRP3 inflammasome activation and TRPV4 blockade data interpretation cast doubts that the data supports certain aspects of the proposed model of signaling downstream of EC-Panx1. These concerns have to be addressed before the manuscript can be considered for publication in Nat. Communications journal.

Major concerns.

1. The suppression of elastin degradation by elastase via Panx1 channel blockade is counter-intuitive and the mechanism is unclear. Did the authors check if PBN treatment can directly suppress the activity of elastase?
 2. The potential mechanism of elastase action on Panx1 channels in endothelial and smooth muscle cells has to be detailed in the description of the model system. My concern is that elastase has been suggested in some studies as an agent that interfere with mechanotransduction (PMIDs: 10831870; 30141981), suggesting possible direct, aneurism-independent activation of Panx1 by this enzyme.
 3. The mechanism of Panx1-mediated activation of NLRP3 inflammasome proposed here is concerning for three reasons: a) Panx1 has been implicated in ATP-mediated and extrinsic apoptosis-mediated activation of NLRP3 inflammasome in different disease models (PMC6517827; accession# 31411729). b) mtDNA release represents the canonical mechanism of AIM2 inflammasome activation, but not NLRP3, as proposed by the authors. As a matter of fact, the activity of NLRP3 inflammasome has been reported as an upstream event leading to mtDNA and DAMPs release (PMC7524928) but not vice versa.
 4. I am concerned with Fig. 6 data interpretation, which resulted in the proposed model of MMP2 activation downstream of the Panx1 –ATP-P2Y-TRPV4 receptor-mediated signaling. The first concern is the TRPV4 blockade data interpretation: despite the effects of P2Y2 and TRPV4 blockade on iCa^{2+} increase and MMP2 production in SMC is very similar, it is unclear why the interpretation of their action on intracellular Ca^{2+} was completely the opposite, ie Ca^{2+} inflow via P2X2 and outflow via TRPV4? Should TRPV4 support the outflow, its blockade by GSK2 would cause the increase rather than a decrease of iCa^{2+} in SMCs.
- My second concern is the proposed model that suggests that purinergic signaling via Panx1-P2Y2 receptor results in a reverse/non-canonical inside-out flow of Ca^{2+} from SMCs into extracellular media via TRPV4 channels. I am not familiar with the reports demonstrating the proposed mechanism since the canonical model of the TRPV4 channel physiology only supports the inflow of extracellular Ca^{2+} . In line with that, the increased channel activation by agonists or by mechanical forces produces cytotoxic influx leading to cell death by calcium overload. The canonical model is in agreement with the fact that the vast majority of physiological and pathological conditions extracellular calcium concentration is much higher (~1000-fold) and the inflow via activated TRPV4 channels is classical and well-characterized activity. In contrast, the non-canonical model proposed in this work claims that TRPV4 opening is capable of releasing intracellular Ca^{2+} against this steep

gradient, the activity that is new to me and has to be clearly demonstrated. In my opinion, this steep gradient thermodynamically precludes a reverse inside-out flow of calcium via an opened channel. Next, even if the outflow occurs, the released calcium puff is unlikely to increase the significantly higher extracellular level and trigger any signal.

In order to agree with the proposed non-canonical model outlined in the Graphical abstract and in Figures 6, 8 of the manuscript, I would like to see experimental data supporting the TRPV4-mediated Ca²⁺ efflux from SMC, which is not obvious from the data in Fig. 6. The authors must provide references and the data on changes in extracellular calcium levels in their cell/tissue culture media allowing them to support the proposed mechanistic model.

5. Human clinical data meta-analysis raises a question regarding pannexin-specificity of the effects of probenecid and spiroolacton treatments in AAA patients. Clinically, spiroolacton is primarily prescribed as aldosterone receptor blocker or as diuretic, which will actively lower blood pressure, as the authors acknowledge in the Discussion along with similar activity reported for probenecid. This activities would directly lower the risk of death from arterial aneurisms through the reduction of arterial pressure, which would be Panx1-independent action. This pannexin1-independent mechanism cannot be discriminated using clinical data but must be discussed.

Minor concerns

1. Page 3, line 108: correct "activate regulate..."
2. PBN treatment reduces the aortic accumulation of ATP at 2 weeks after elastase treatment and 4 weeks in the AngII-induced model. This is despite the peak release of ATP from ECs is observed at 12h post-elastase treatment. Since ATP is very labile and is rapidly converted to adenosine, the peak Panx1-mediated ATP release via Panx1 can most likely be captured at the earlier vs later data points, which should also be considered to understand the dynamics of the release.
3. Page 9, line 236 "... by GSK2 or P2X4 inhibition..." needs correction: GSK2 is a drug, not a receptor.
4. Page 9, lines 247-248 contain a statement not supported by the data "...Panx1/ATP release activates P2X4 receptor..". A similar unsupported statement is on line 290: "SMC activation via P2X4Rs.."
5. In the discussion (lines 351-353) the sentence starting with " This data..." is rather open-ended and redundant to the last paragraph. Suggest to re-write to improve the flow

Reviewer #2 (Remarks to the Author):

General/conceptual:

This study investigates role of Panx1 on endothelial cells in aortic inflammation and remodelling during AAA development. The basic issue in this paper is that the authors are trying to establish a link between Panx1 channels and ATP release. What was the rationale then behind measuring TOTAL ATP level (mostly reflecting intracellular ATP) in aortic tissue homogenates? They also used a lot of exogenous ATP for in vitro treatments - 1 mM. This amount is non-physiological. Leukocyte trafficking was also investigated. Extracellular ATP metabolism and especially adenosine (not ATP as such) in this cascade is intimately involved in controlling leukocyte trafficking. How was this taken into account?

Specific comments:

Figure 1, D-H, J; Fig. 2 E, G; 3H, 4 E, G, H: the statistical differences are presented in a very strange way. Please, take ns away and indicate the statistically significant comparisons as they are. Now the reader is left to think that you are for example comparing TNF-a and MCP-1 within EC Panx1^{-/-} mice.

Figures 2 and 3D, 4D need quantification. The immunohistochemical stainings can be left as examples.

Figure 5D: What are the numbers on the y-axis? Number of cells? Percentage of input cells??

Page 9: The authors write: These results demonstrate that EC-induced SMC activation is mediated by EC-specific Panx1/ATP release that activates P2X4 receptors on SMCs to increase intracellular Ca²⁺, that is released via TRPV4 channels (Figure 6F). Based on the previous sentences and Figure 6F, should it be P2Y2?

Materials and methods: Human AAA tissue and clinical data analysis paragraph includes statistical analyses, although there is a separate Statistics section. Please, put all statistics under the heading Statistics.

Minor comments:

Line 468: please define ectonucleotidase as there are several ones of those (ARL67156 is CD39 inhibitor)

Line 450: tissue were

Line 503: transfer were performed

Line 454: be consistent; the catalog number is given, in most cases not

Reviewer #3 (Remarks to the Author):

This interesting animal study examines the effect of Pannexin-1 (Panx1) channels in AAA models. The impact of the study would be increased in my opinion if:

1. The methods of the animal studies are a little concerning. One of the issues raised with prior animal studies is the lack of use of methods to minimize bias. This includes randomization of animals, blinding of investigators and particular outcome assessors, well justified sample sizes and ITT analyses. Placebo controlled is relevant to the drug studies. The sample sizes for most of the studies are very small 5-6 seems to be mentioned and this appears far too small for reliable validation. There is limited detail that the biases of many animal experiments have been dealt with. If this is the case this should be made very clear. Pictures of all aortas should be provided in the supplement not just examples.

2. The authors provide human data which is good but much more needs to be understood about this data. What was the aneurysm size and growth. The authors present data on mortality and repair. This presumably is all cause mortality and repair rates are very low 5% and no different between groups. This implies these must be patients with small AAAs and aneurysm rupture is therefore not going to be the cause of death. Thus the association with the drugs indicated is not likely related to AAA but other causes of death. This needs to be examined in much greater detail and if relevant needs to be presented in the main paper or dropped. Is the channel really the main target of these drugs in any case? If so please provide evidence to support at the dose taken they really effect these channels as opposed to other targets.

Other points

1. Please provide details of how aortic diameter was measured and reproducibility statistics for this.

2. The drug study in the animals could be designed to randomize and administer the drug/ placebo well after aneurysm are established similar to what is required in patients.

3. I could not see how the statistical analysis was performed in the animal studies- please provide.

Response to Reviewers

We thank reviewers for the supporting feedback and suggestions. Kindly see our point-by-point response to all queries below. We have revised our manuscript accordingly, which now includes new figures and data. Page numbers below refer to the marked version of the revised manuscript and the tracked changes are highlighted in red color in the marked version.

Reviewer #1 (Remarks to the Author):

The manuscript by Filiberto and co-authors investigates the role of endothelial Panx1 in aortic inflammation and remodeling during abdominal aortic aneurysms development. The high mortality rate among patients with AAA represents an urgent and unaddressed clinical problem with a low success rate of surgical intervention as the only treatment option. An urgent need for new preventive therapies targeting inflammatory and proteolytic pathways in the aorta underscores the clinical relevance of this research.

The authors use two animal models of AAA and perform in vitro experiments to define the role of Panx1 in AAA pathology, and define its downstream partners in ECs and SMCs that are critical players for neutrophil transmigration, macrophage activation, SMC activation and remodeling. The study tests a hypothesis that Panx1 plays a key mechanistic role in the regulation of these pathological pathways and can serve as a target to alleviate AAA progression. Observations performed in vivo are dissected down to distinct cell-type-specific interactions. Another line of evidence supporting their hypothesis is produced by retrospective meta-analysis of clinical data in human patients with AAA who received off-label treatments with two drugs that have Panx1-inhibitory activity.

Overall, this is a well-designed and written manuscript. However, several major and minor concerns dealing with data potential indirect effects of elastase on Panx1 activity proposed mechanisms of NLRP3 inflammasome activation and TRPV4 blockade data interpretation cast doubts that the data supports certain aspects of the proposed model of signaling downstream of EC-Panx1. These concerns have to be addressed before the manuscript can be considered for publication in Nat. Communications journal.

(Major concerns)

Comment 1. The suppression of elastin degradation by elastase via Panx1 channel blockade is counter-intuitive and the mechanism is unclear. Did the authors check if PBN treatment can directly suppress the activity of elastase?

Response: It has been previously established by our group and others that the rate of elastin degradation and fragmentation is accelerated under clinical and experimental conditions such as abdominal aortic aneurysms (**PMIDs:** 3649236, 17372168, 23413358 and 22965992). The destructive changes in structural integrity of elastin in AAA are secondary to chronic inflammatory conditions in the surrounding aortic layers. The pancreatic porcine elastase acts as a serine proteinase and sets off a cascade of inflammation and vascular remodeling in our transient topical elastase model (**PMIDs:** 32506673, 24030402). We did not check the direct suppression of elastase activity as the well-established model of using elastase treatment is transient (5 minutes followed by wash off) and we were interested to delineate how the Pannexin-1 (Panx1) channels affect the downstream signaling cascade of aneurysm formation. Also, we used a second established model of angiotensin II in ApoE^{-/-} mice (devoid of any elastase treatment) which also displays elastin degradation, that also displayed inhibition of aneurysm formation after Probenecid treatment. Collectively, our studies were conducted using two established experimental murine models and provide supportive evidence for panx1 modulation by probenecid treatment for the immune regulation of aortic inflammation and vascular remodeling in AAA formation.

Comment 2. The potential mechanism of elastase action on Panx1 channels in endothelial and smooth muscle cells has to be detailed in the description of the model system. My concern is that elastase has been suggested in some studies as an agent that interfere with mechanotransduction (**PMIDs:** 10831870; 30141981), suggesting possible direct, aneurism-independent activation of Panx1 by this enzyme.

Response: This is actually a supportive argument for our model as mechanotransduction is an intricate part of aortic aneurysm formation due to mechanical stress forces (PMID: 32936305). Therefore, if elastase treatment induces mechanotransduction effects in our experimental model, then this would be clinically relevant for the process of AAA formation which involves shear stress forces during vascular remodeling. Also, Panx1 channels have been independently known to be induced by mechanotransduction forces (PMIDs: 32936305, 24048216 and 23070703) and ours is the first study to decipher the correlation between initial sensors to hemodynamics forces in blood vessels with G-protein-coupled receptors (GPCRs), and relevant ion channels i.e. Panx1 and TRPV4. The molecular mechanotransduction events in the vessel wall triggered by disturbed shear stress on vascular endothelial cells, and cyclic stretch in endothelial cells and vascular smooth muscle cells plays a pivotal role in vascular remodeling during AAA formation. Further support for Panx1 channels in AAA, independent of elastase-induced mechanotransduction, is provided by our second murine model of angiotensin II induction of AAA that does not involve elastase, and still displays a marked attenuation of aneurysm formation after Probenecid treatment. Taken together, these data confirm a significant and novel role of Panx1 channels in vascular inflammation and aortic aneurysm pathogenesis.

Comment 3. The mechanism of Panx1-mediated activation of NLRP3 inflammasome proposed here is concerning for three reasons: a) Panx1 has been implicated in ATP-mediated and extrinsic apoptosis-mediated activation of NLRP3 inflammasome in different disease models (PMC6517827; accession# 31411729). b) mtDNA release represents the canonical mechanism of AIM2 inflammasome activation, but not NLRP3, as proposed by the authors. As a matter of fact, the activity of NLRP3 inflammasome has been reported as an upstream event leading to mtDNA and DAMPs release (PMC7524928) but not vice versa.

Response: The regulatory mechanisms of IL-1 β secretion have been shown to be driven by varied mechanisms. Although NLRP3 inflammasome can activate mtDNA but multiple recent reports have also shown the vice versa to occur as well. Previous studies have shown the role of mitochondrial DNA (mtDNA) in IL-1 β secretion through NLRP3 inflammasome activation (PMIDs: 34306320, 32009974, 31903272, 24849809). Cellular localization of and access to mtDNA are critical factors in determining whether NLRP3 or AIM2 are activated for IL-1 β release as Shimada *et al* have shown that oxidized DNA can induce inflammasome activation via preferential activation of NLRP3 and not AIM2 (PMID: 22342844). Furthermore, Nakahira *et al* have shown that mtDNA can serve as an AIM2-independent co-activator of caspase-1 activity in conjunction with LPS and ATP. Also, extracellular ATP activates the NLRP3 inflammasome by provoking intracellular potassium efflux and mitochondrial DNA release into cytoplasm (PMID: 24849809). Purinergic P2X7 receptor stimulation also induces fast release into the cytosol of oxidized mtDNA that promotes NLRP3 inflammasome assembly by direct interaction (PMID: 21151103 and 22342844). Taken together, these results are supportive of our data where Panx1-mediated ATP activates P2X7R on macrophages to stimulate IL-1 β secretion via mtDNA release.

Comment 4. I am concerned with Fig. 6 data interpretation, which resulted in the proposed model of MMP2 activation downstream of the Panx1 –ATP-P2Y-TRPV4 receptor-mediated signaling. The first concern is the TRPV4 blockade data interpretation: despite the effects of P2Y2 and TRPV4 blockade on iCa $^{2+}$ increase and MMP2 production in SMC is very similar, it is unclear why the interpretation of their action on intracellular Ca $^{2+}$ was completely the opposite, i.e. Ca $^{2+}$ inflow via P2X2 and outflow via TRPV4? Should TRPV4 support the outflow, its blockade by GSK2 would cause the increase rather than a decrease of iCa $^{2+}$ in SMCs.

My second concern is the proposed model that suggests that purinergic signaling via Panx1-P2Y2 receptor results in a reverse/non-canonical inside-out flow of Ca $^{2+}$ from SMCs into extracellular media via TRPV4 channels. I am not familiar with the reports demonstrating the proposed mechanism since the canonical model of the TRPV4 channel physiology only supports the inflow of extracellular Ca $^{2+}$. In line with that, the increased channel activation by agonists or by mechanical forces produces cytotoxic influx leading to cell death by calcium overload. The canonical model is in agreement with the fact that the vast majority of physiological and pathological conditions extracellular calcium concentration is

much higher (~1000-fold) and the inflow via activated TRPV4 channels is classical and well-characterized activity. In contrast, the non-canonical model proposed in this work claims that TRPV4 opening is capable of releasing intracellular Ca^{2+} against this steep gradient, the activity that is new to me and has to be clearly demonstrated. In my opinion, this steep gradient thermodynamically precludes a reverse inside-out flow of calcium via an opened channel. Next, even if the outflow occurs, the released calcium puff is unlikely to increase the significantly higher extracellular level and trigger any signal.

In order to agree with the proposed non-canonical model outlined in the Graphical abstract and in Figures 6, 8 of the manuscript, I would like to see experimental data supporting the TRPV4-mediated Ca^{2+} efflux from SMC, which is not obvious from the data in Fig. 6. The authors must provide references and the data on changes in extracellular calcium levels in their cell/tissue culture media allowing them to support the proposed mechanistic model.

Response: We apologize for the confusion in the displayed schematic which has now been rectified (please see new **Fig. 6f** in revised manuscript). Our data is clear on the proposed interactions between Panx1-ATP-P2Y2-TRPV4 receptor mediated signaling in aortic smooth muscle cells (SMCs) where an **influx** (not efflux) of intracellular Ca^{2+} occurs downstream of TRPV4 activation. The efflux of Ca^{2+} in the previous schematic was suggested as TRPV4 and NCX ($\text{Na}^+/\text{Ca}^{2+}$ exchanger) have been shown to physically and functionally interact on SMCs (PMID: 31866874) which can cause Ca^{2+} efflux (please see **Fig. 1** below). However, since we demonstrate only the TRPV4 mediated increase in iCa^{2+} leading to upregulation of MMP2 activity, but not the efflux via $\text{Na}^+/\text{Ca}^{2+}$ exchanger, therefore we have now revised the schematic to reflect our findings. Furthermore, **new data** (**Fig. 6a** in revised manuscript) has now been provided where treatment with a TRPV4 agonist, GSK1016790A (GSK1; 20 nM; Tocris) stimulates intracellular Ca^{2+} increase in SMCs, further clarifying the involvement of the canonical pathway in our model.

To further clarify, the conceptual design of this experiment was created to decipher how Panx1 on endothelial cells are involved in initiation and propagation of intercellular calcium waves in the neighboring aortic SMCs. Mechanical stress (or other stimuli, like depolarization) during vascular remodeling of aortic tissue opens Panx1 channels, enabling the exit of cytoplasmic ATP. Extracellular ATP activates purinergic P2Y2 receptors on cells within diffusion distance i.e. SMCs. P2Y2 receptors are G-protein coupled and activate phospholipase C resulting in IP_3 release and increased calcium from intracellular stores (supported by a recent study by our co-author Dr. Isakson; **Fig. 2**, please see above; PMID: 34490843) but also is free to permeate gap junction channels to contiguous cells. The transient increase in cytoplasmic calcium concentration following P2Y2 stimulation can sensitize TRPV4 channels, resulting in an increase in influx of extracellular calcium. Efflux of Ca^{2+} can be mediated through $\text{Na}^+/\text{Ca}^{2+}$ exchanger, which has been shown to physically and functionally interact

with TRPV4 channels (**Fig. 1**, please see above), and alterations in Ca^{2+} homeostasis can modulate MMP2 activity (**PMID: 8063786**). This cascade leads to increase in MMP2 activity leading to SMC degradation and subsequent vascular remodeling. This description has now been added to the revised manuscript for added clarification (pages 12-13, line 298-310).

We have now rectified the error and the schematic has been appropriately revised for clarity which now addresses the first concern raised by the reviewer. As the data in **Fig. 6A** in revised manuscript suggests, TRPV4 blockade by GSK2 (TRPV4 antagonist) causes decrease in intracellular (i) Ca^{2+} in SMCs while stimulation of TRPV4 by GSK1 (TRPV4 agonist) increases iCa^{2+} in SMCs. The second concern raised by the reviewer is also resolved now as we are proposing an increase in iCa^{2+} downstream of purinergic signaling via Panx1-P2Y2-TRPV4 signaling based on our data.

Comment 5. Human clinical data meta-analysis raises a question regarding pannexin-specificity of the effects of probenecid and spiroolacton treatments in AAA patients. Clinically, spiroolacton is primarily prescribed as aldosterone receptor blocker or as diuretic, which will actively lower blood pressure, as the authors acknowledge in the Discussion along with similar activity reported for probenecid. This activities would directly lower the risk of death from arterial aneurysms through the reduction of arterial pressure, which would be Panx1-independent action. This pannexin1-independent mechanism cannot be discriminated using clinical data but must be discussed.

Response: We thank the reviewer for bringing up an excellent point. Previous studies from our group and others have confirmed the Pannexin-1 (Panx1) inhibition specificity of Probenecid and Spironolactone (**PMID:29084879, 21546608, 29237722, 18596212, 29222419, 29745255**) and the experimental data in this manuscript is the first report to show the role of Panx1 signaling and relevant pharmacological compounds to immunomodulate AAA formation. Established literature from our group and others also suggest that pannexin 1/ATP signaling pathway can participate in the regulation of vascular tone and blood pressure (**PMID: 25921414, 32446934, 30026274, 29449359**). Therefore the reduction of arterial pressure is a Panx1-dependent action that is plausibly associated with the decrease in aneurysm formation and subsequent risk of death from aneurysm rupture. Our data is the first experimental and clinical report to show this previously undescribed association in the vascular pathogenesis of aortic aneurysms. While our clinical meta-analysis does not decipher between the direct versus indirect effects of Panx1 specific inhibitors in reducing mortality of AAA patients, but it provides associative evidence of a novel therapeutic option by the use of repurposed medications like Probenecid and Spironolactone in treating these patients. The clinical relevance of these inhibitors via mechanosensitive modulation of Panx1 channels in aortic tissue and/or by regulation of reduction of arterial pressure remains to be deciphered in the patient population. These limitations pertaining to the clinical data interpretation have now been further discussed with appropriate references in the Discussion segment of the manuscript (**page 13, line 322-331**).

Minor concerns:

Comment 6. Page 3, line 108: correct “activate regulate...”

Response: This error has now been corrected (**page 3, line 87**).

Comment 7. PBN treatment reduces the aortic accumulation of ATP at 2 weeks after elastase treatment and 4 weeks in the AngII-induced model. This is despite the peak release of ATP from ECs is observed at 12h post-elastase treatment. Since ATP is very labile and is rapidly converted to adenosine, the peak Panx1-mediated ATP release via Panx1 can most likely be captured at the earlier vs later data points, which should also be considered to understand the dynamics of the release.

Response: Thank you for bringing up this suggestion. We have previously analyzed the ATP content in aortic tissue at days 3 and 7 (in addition to day 14) post-elastase treatment with and without Probenecid treatment. The elastase-induced increase in ATP release was attenuated by Probenecid treatment on days 3, 7 and 14 compared to untreated controls. This **new data (Fig. 3o** in revised manuscript) has now be included in the revised manuscript.

Comment 8. Page 9, line 236 “... by GSK2 or P2X4 inhibition...” needs correction: GSK2 is a drug, not a receptor.

Response: The error has now been rectified (**page 8, line 198**).

Comment 9. Page 9, lines 247-248 contain a statement not supported by the data "...Panx1/ATP release activates P2X4 receptor..". A similar unsupported statement is on line 290: "SMC activation via P2X4Rs.."

Response: We apologize for the oversight. The correct P2Y2 receptor has now replaced the erroneous P2X4 description (**page 8, line 212; and page 10, line 247**).

Comment 10. In the discussion (lines 351-353) the sentence starting with " This data...." is rather open-ended and redundant to the last paragraph. Suggest to re-write to improve the flow

Response: The sentence has been edited in the revised version (**pages 13-14, lines 322-331**).

Reviewer #2 (Remarks to the Author):

General/conceptual:

Comment 1: This study investigates role of Panx1 on endothelial cells in aortic inflammation and remodelling during AAA development. The basic issue in this paper is that the authors are trying to establish a link between Panx1 channels and ATP release. What was the rationale then behind measuring TOTAL ATP level (mostly reflecting intracellular ATP) in aortic tissue homogenates? They also used a lot of exogenous ATP for in vitro treatments - 1 mM. This amount is non-physiological. Leukocyte trafficking was also investigated. Extracellular ATP metabolism and especially adenosine (not ATP as such) in this cascade is intimately involved in controlling leukocyte trafficking. How was this taken into account?

Response: Recent studies from our group of authors have shown a critical role of Pannexin-1 (Panx1) signaling in various cellular processes like apoptosis, tissue inflammation and blood pressure homeostasis (*Nature*, 2020, 580(7801):130-135 (**PMID**: 32238926); *Circulation Research*, 2018, 122(4):606-615 (**PMID**: 29237722); *Nature Communications*, 2015 Aug 5;6:7965 (**PMID**: 26242575); *Nature*, 2014, 507(7492):329-34 (**PMID**: 24646995); *Nature*, 2010, 467(7317):863-7 (**PMID**: 20944749); *Nature*, 2009; 61(7261):282-6 (**PMID**: 19741708). The current study focuses on a clinical disease process i.e. aortic aneurysms, in which the contribution of Panx1 signaling has not yet been described. Therefore, we deciphered the mechanistic signaling of Panx1 and determined the repurposed use of a known pharmacological compound, Probenecid (currently used in the treatment of gout), to affect experimental and clinical outcomes of AAA.

We have previously well established that Panx1 channels act as a conduit for ATP release (**PMIDs**: 20944749, 22311983, 32446934, 29745255, 29237722, 26242575, 22311983, 28134257). In this study, we measured total ATP in aortic tissue homogenates to specifically demonstrate the elevated expression of this nucleotide in the conserved section of aneurysm formation of the blood vessel which involves significant inflammation and vascular remodeling. We also measured sequential expression of extracellular ATP in plasma of these animals in our experimental model and this **new data (Supplementary Fig. 2** in revised manuscript) will now be included in the revised manuscript. The multi-fold increase of extracellular ATP in plasma of elastase-treated mice was significantly attenuated by Probenecid treatment, further confirming our findings. Furthermore, we have now rectified the oversight in displaying the dose of recombinant ATP which was used as 1 μ M (not 1mM; symbol font has been corrected now in the text; **page 20, line 487**). This physiological dose was selected to illicit an optimal response in cultured SMCs based on previous studies (**PMID**: 10070169, 34490843)

Metabolism of endogenous ATP to adenosine can result in activation of various adenosine receptors (A1, A2A, A2B and/or A3). We have previously shown that A_{2A}R activation attenuates AAA formation partly by inhibiting immune cell recruitment and reducing elastin fragmentation (**PMID**: 23413358). However, the role of other adenosine receptors is unclear in AAA formation. It is plausible that adenosine receptors, especially A_{2B}AR, can be proinflammatory that can negate the protective effects of endogenous adenosine/A_{2A}AR activation. As ATP is an upstream DAMP molecule so excessive secretion of extracellular ATP can cause the dysregulation of the ATP-ADP-AMP-adenosine

pathway to skew the local tissue milieu towards inflammation and purinergic signaling mediated cell death rather than adenosine mediated cell survival, as recently reported by our co-author (Dr. Isakson; **Fig. 3**, see below; **PMID**: 29866797).

Specific comments:

Comment 2. Figure 1, D-H, J; Fig. 2 E, G; 3H, 4 E, G, H: the statistical differences are presented in a very strange way. Please, take ns away and indicate the statistically significant comparisons as they are. Now the reader is left to think that you are for example comparing TNF- α and MCP-1 within EC Panx1^{-/-} mice.

Response: We thank the reviewer for this suggestion and relevant statistical comparisons have now be displayed for **Figs. 1-3** in revised manuscript.

Comment 3. Figures 2 and 3D, 4D need quantification. The immunohistochemical stainings can be left as examples.

Response: We have now quantified the IHC staining's in Figs. 2, 3D and 4D by independent observers. Images were acquired using AxioCam Software version 4.6 and an AxioCam MRc camera (Carl Zeiss Inc., Thornwood, New York). Threshold gated positive signal was detected within the AOI and quantified using Image-Pro Plus version 7.0 (Media Cybernetics Inc., Bethesda, Maryland). Elastin degradation was quantified by counting the number of breaks per vessel and then averaged and graphed. These new details (page 19, line 442-445) and data (Figs. 2b-f, 3e-i, and 4e-i in revised manuscript) have now be provided which has relative quantifications of immunohistochemical staining in addition to the previous qualitative displays of representative images.

Comment 4. Figure 5D: What are the numbers on the y-axis? Number of cells? Percentage of input cells??

Response: The y-axis in Fig. 5D represents number Relative Fluorescence Units as a measure of fluorescently-labeled neutrophils that transmigrated through the endothelial layers. The y-axis has now been revised to display 'Neutrophil numbers (Relative Fluorescence Units)' in **Fig. 5d** in revised manuscript.

Comment 5. Page 9: Thee authors write: These results demonstrate that EC-induced SMC activation is mediated by EC-specific Panx1/ATP release that activates P2X4 receptors on SMCs to increase intracellular Ca²⁺, that is released via TRPV4 channels (Figure 6F). Based on the previous sentences and Figure 6F, should it be P2Y2?

Results: We apologize for the oversight and have now corrected the text to represent P2Y2 (**page 8, line 212**).

Comment 6. Matrials and methods: Human AAA tissue and clinical data analysis paragraph includes statistical analyses, although there is a separate Statistics section. Please, put all statistics under the heading Statistics.

Results: We have now included all statistical analyses under **Methods section (page 22, line 519-532)**.

Minor comments:

Line 468: please define ectonucleotidase as there are several ones of those (ARL67156 is CD39

[REDACTED]

inhibitor)

Line 450: tissue were

Line 503: transfer were performed

Line 454: be consistent; the catalog number is given, in most cases not

Responses to minor comments: ARL67156 has been defined as Ectonucleoside triphosphate diphosphohydrolase-1 (CD39/NTPDase 1) in the text (**page 19; line 464**). All other suggested edits mentioned above have been now incorporated in the text (**page 19, line 446; page 21, line 501; page 19, line 450**).

Reviewer #3 (Remarks to the Author):

This interesting animal study examines the effect of Pannexin-1 (Panx1) channels in AAA models. The impact of the study would be increased in my opinion if:

Comment 1. The methods of the animal studies are a little concerning. One of the issues raised with prior animal studies is the lack of use of methods to minimize bias. This includes randomization of animals, blinding of investigators and particular outcome assessors, well justified sample sizes and ITT analyses. Placebo controlled is relevant to the drug studies. The sample sizes for most of the studies are very small 5-6 seems to be mentioned and this appears far too small for reliable validation. There is limited detail that the biases of many animal experiments have been dealt with. If this is the case this should be made very clear. Pictures of all aortas should be provided in the supplement not just examples.

Response: Throughout our study, we ensured scientific rigor, integrity, robust and unbiased approach through several mechanisms. All experiments started from a testable hypothesis or prediction and experiments were planned without bias on the outcome and performed in a blinded manner. Our mice were electronically tagged to facilitate with the blinding process. To ensure randomization, we used an I.D. code in conjunction with our in house scripts to assign animals prior to surgery to treatment groups, and this number was used for all samples generated from these animals which were unblinded at the end of each experimental series. Outcome assessment and scoring of animals for all samples was performed using automated software or by blinded observers. For microscopy, we used independent observers using random field selection and image analysis. We utilized highly controlled conditions to maximize reproducibility and experiments were repeated several times with varying conditions to ensure robustness. These details have now been included in the methodology section of the revised manuscript (**Methods section; page 16, line 383-391**).

For our mouse phenotype experiments, we used n=11-13 per group which is an appropriate sample size for power analysis performed in consultation with our Biostatistics core. The mouse aortic tissue is very small in size and can be used for analyses of protein expression or immunohistochemistry. Therefore, we utilized n=5-6/group (used in triplicates for respective analyses) which showed reliable and reproducible results.

Comment 2. The authors provide human data which is good but much more needs to be understood about this data. What was the aneurysm size and growth. The authors present data on mortality and repair. This presumably is all cause mortality and repair rates are very low 5% and no different between groups. This implies these must be patients with small AAAs and aneurysm rupture is therefore not going to be the cause of death. Thus the association with the drugs indicated is not likely related to AAA but other causes of death. This needs to be examined in much greater detail and if relevant needs to be presented in the main paper or dropped. Is the channel really the main target of these drugs in any case? If so please provide evidence to support at the dose taken they really effect these channels as opposed to other targets.

Response: We agree that the retrospective clinical meta-analyses does not conclusively determine a direct correlation between pannexin inhibitors and a decreased incidence of aneurysm size or rupture-associated mortality. As discussed in the text (line 349-353), this data is the first report to characterize and demonstrate an associative correlation for an overall decreased mortality in AAA patients that were

taking these medications compared to untreated patients. Previous studies from our group and others have confirmed the Panx1 inhibition specificity of Probenecid and Spironolactone (PMIDs: 29084879, 21546608, 29237722, 18596212, 29222419, 29745255) and the experimental data in this manuscript is the first report to show the role of Panx1 signaling and relevant pharmacological compounds to immunomodulate AAA formation. The novelty of this data underscores the previously undescribed importance of Panx1 inhibitors for patients with aortic aneurysms and opens the door for further detailed clinical studies for describing aneurysm size and aortic rupture to establish a conclusive association.

Other points:

Comment 3. Please provide details of how aortic diameter was measured and reproducibility statistics for this.

Response: Aortic diameters were measured by video micrometry using NIS-Elements D5.10.01 software attached to the microscope (Nikon SMZ-25; Nikon Instruments, Melville, NY). Aortic dilation percentage was determined by $[(\text{maximal AAA diameter} - \text{self-control aortic diameter}) / (\text{self-control aortic diameter})] \times 100$. Aortic dilation of $\geq 100\%$ was considered positive for AAA. These details have now been added to the methods section (page 18; line 421-425).

Comment 4. The drug study in the animals could be designed to randomize and administer the drug/ placebo well after aneurysm are established similar to what is required in patients.

Response: Thank you for this suggestion. We have now included new data (Supplementary Fig. 3 in revised manuscript) showing that administration of Probenecid after AAA formation attenuates aortic diameter compared to untreated controls. These clinically relevant findings provide supportive evidence for translational significance.

Comment 5. I could not see how the statistical analysis was performed in the animal studies- please provide.

Response: Statistical analysis for animal and human studies is provided in the **Methods section (page 22, line 519-532).**

REVIEWER COMMENTS

Reviewer #1 (Remarks to the Author):

In the revised manuscript, Filiberto and co-authors have addressed most deficiencies and concerns, however, I have one concern that has not been properly addressed. Therefore, the manuscript requires a minor revision.

The new data with the TRPV4 blockade allowed the revision of figure 6a, which is now depicting Ca²⁺ inflow via TRPV4. Despite this major improvement, I am still concerned with the confusing schematics in Fig. 6f, containing the author's interpretation of the data: combined with the description in the text, it appears as showing a Ca²⁺ efflux is leading to MMP activation. The authors cite a recent paper where TRPV4-mediated Ca²⁺ influx was shown to be modulated via the NCX exchanger to maintain contractility, which does not show a massive Ca²⁺ efflux, but rather refers to its "safety valve" function to preserve intracellular calcium homeostasis. In my opinion, even a concerted release from ER stores, which can significantly increase intracellular concentration, is very unlikely to alter the extracellular one to activate MMPs.

It should be clarified both in the text and in the schematic that, according to the author's conclusion, "destabilization of intracellular Ca²⁺ homeostasis" rather than a massive release of Ca²⁺ via NCX facilitates MMP2 production.

Reviewer #2 (Remarks to the Author):

The manuscript is now fine, but information regarding how the plasma was collected for ATP analyses (what type of plasma, EDTA???) needs to be included as well as real concentrations of ATP.

Reviewer #3 (Remarks to the Author):

No Further comments

Response to Reviewers

We thank the reviewers for the positive feedback of our study. The remaining suggestions have now been incorporated in the manuscript. Page numbers below refer to the marked version of the revised manuscript and the tracked changes are highlighted in red color in the marked version.

Reviewer #1 (Remarks to the Author):

In the revised manuscript, Filiberto and co-authors have addressed most deficiencies and concerns, however, I have one concern that has not been properly addressed. Therefore, the manuscript requires a minor revision.

The new data with the TRPV4 blockade allowed the revision of figure 6a, which is now depicting Ca²⁺ inflow via TRPV4. Despite this major improvement, I am still concerned with the confusing schematics in Fig. 6f, containing the author's interpretation of the data: combined with the description in the text, it appears as showing a Ca²⁺ efflux is leading to MMP activation. The authors cite a recent paper where TRPV4-mediated Ca²⁺ influx was shown to be modulated via the NCX exchanger to maintain contractility, which does not show a massive Ca²⁺ efflux, but rather refers to its "safety valve" function to preserve intracellular calcium homeostasis. In my opinion, even a concerted release from ER stores, which can significantly increase intracellular concentration, is very unlikely to alter the extracellular one to activate MMPs.

It should be clarified both in the text and in the schematic that, according to the author's conclusion, "destabilization of intracellular Ca²⁺ homeostasis" rather than a massive release of Ca²⁺ via NCX facilitates MMP2 production.

Response: We thank the reviewer for this suggestion and have now edited the statement in the text (Page 13) and incorporated the relevant adjustments to the schematic (Fig. 6f) and respective legend (Page 31). The clarification in the text and schematic now reiterates that destabilization of intracellular Ca²⁺ homeostasis facilitates increased MMP2 activity leading to vascular remodeling during AAA formation.

Reviewer #2 (Remarks to the Author):

The manuscript is now fine, but information regarding how the plasma was collected for ATP analyses (what type of plasma, EDTA???) needs to be included as well as real concentrations of ATP.

Response: We have now included the relevant details of plasma collection in the Methods section (Page 18) as well as displayed the concentration values of eATP (Supplementary Fig. 2).

REVIEWERS' COMMENTS

Reviewer #1 (Remarks to the Author):

Thank you for addressing all concerns and recommendations.

Reviewer #2 (Remarks to the Author):

The authors have appropriately responded to my criticism.

Response to Reviewers:

We thank all the reviewers for the helpful suggestions and recommendations.

REVIEWERS' COMMENTS

Reviewer #1 (Remarks to the Author):

Thank you for addressing all concerns and recommendations.

Reviewer #2 (Remarks to the Author):

The authors have appropriately responded to my criticism.